# Manipulating mechanical strength of isoreticular two-dimensional polyamide materials via multiple interactions

Qing Hu[1], Chiran Wang[1], Yuan Zhao[1], Awei Hu [2] & Bo Liu [1,2] ✉

Anisotropic two-dimensional (2D) materials exhibit huge difference in the directions parallel and vertical to the plane, especially in mechanical properties, and the connecting interactions among 2D nanosheets dominates the bulk mechanical strength owing to the challenge to prepare continuous, single-crystal 2D material film at macroscopic scale. Herein, we report a series of isoreticular 2D polyamide materials and reveal that smaller structural units result in higher Young's modulus, while multiple weak forces including the hydrogen-bond, π-π and mismatched electrostatic interactions endows the bulk 2D materials superior elasticity, hardness and yield strength. Specifically, The Young's modulus and hardness of 2D polyamide film (GH-TMC) reach 35.6 GPa and 2.0 GPa, and the elastic recovery rate is as high as 60%, overwhelming the most polymer, metal and metal-organic framework (MOF)/ covalent organic framework (COF) materials. The synergy of rigid small-ring units, high-density H-bond networks, and π-π/electrostatic interactions enable these films to bridge the gap between inorganic and polymeric materials, making them ideal for flexible electronics, high-performance protective coatings, and energy devices. The strategy via designing molecular structure and interactions among nanosheets in 2D materials enable us fabricate super 2D materials with overall mechanical strength.

Two-dimensional (2D) materials have sparked significant research interests in material science since the discovery of graphene[1], owing to the atomic-thick structure and potential applications across various fields, including energy storage[2], catalysis[3], coating and protection[4], electronics[5] and nano-mechanics[6]. As the basic properties of 2D materials, strength and elasticity affect their stability and durability in various applications[7]. Owing to the pronounced anisotropic properties of 2D materials, the Young's modulus, indicating the in-plane strength by measuring the strain against vertical stress, is usually high, while the tensile modulus and elasticity reflecting the strain against stress along the plane is relatively low, because the interlayer interaction is weak and most 2D materials comprise of the stacking nanosheets at macroscopic scale[8–10]. The mechanical strength of 2D materials is governed by the nature of covalent bonding, the rigidity of constituent units, bond density, and in-plane porosity characteristics[11–13]. Inorganic 2D materials such as graphene, hexagonal boron nitride, molybdenum disulfide ($MoS_2$), MXenes and graphdiyne usually exhibit high Young's modulus in a range of 100–1000 GPa but less structural tunability[14–18]. Graphene, with a rigid and dense conjugated six-membered ring comprising of alternative single and double C-C bonds, achieves the highest modulus of 1 TPa due to its structural feature[19].

In contrast, 2D organic polymer materials exhibit enriched structural tailorability and tunability, as well as high elasticity, but low Young's modulus, typically in the range of 1–10 GPa[20–22]. 2D covalent organic frameworks (COFs), typically assembled from rigid motifs form conjugated structures and stacked through π-π interactions with

[1]School of Chemistry and Materials Science, University of Science and Technology of China, Hefei, Anhui, China. [2]Hefei National Research Center for Physical Sciences at the Microscale, University of Science and Technology of China, Hefei, Anhui, China. ✉e-mail: liuchem@ustc.edu.cn

low bond density, exhibits weak mechanical strength[23]. Kevlar, as a famous polyamide, exhibits high modulus owing to massive hydrogen-bond (H-bond) interaction among polyamide backbones[24]. For 2D materials, the stacking modes of monolayers and intralayer interactions greatly affect their modulus[25]. Essentially, preparation of bulk 2D materials at macroscopic scales remains challenging due to their inherent anisotropy[26]. Strong in-plane covalent bonds contrast with weak interlayer interactions (van der Waals, π-π), leading to preferential nanosheet stacking rather than monolithic single-crystal formation[27–29]. For instance, even graphene's modulus drops significantly (from ~1 TPa to 27–478 GPa) when composed of stacked nanosheets[30,31]. In addition, achieving both crystallinity and processability requires balancing competing interactions (H-bonds, π-π stacking, and electrostatic forces)[32,33]. These factors collectively limit the scalability of defect-free, continuous 2D material films.

The pursuit of high mechanical strength in 2D polymers is critical because their ultrathin architecture typically compromises robustness[34]. Most 2D materials exhibit brittleness or poor elasticity, limiting real-world use[35]. Our work overcomes this problem by reducing the size of rigid structural units and the synergistic effects of multiple weak interactions (H-bonds, π-π stacking, electrostatic interactions) to achieve an unprecedented modulus and hardness, while maintaining polymer-like flexibility and being comparable to metals. Such materials are urgently needed in flexible bioelectronics for strain-insensitive health monitoring[36], ultra-sensitive gas sensing[37] and high-performance electronic devices[38], which mechanical integrity dictates device lifespan.

In this work, a series of 2D ring-structured polyamides featuring distinct ring sizes, including GH-TMC, GH-BTCA, GH-OC, Melem-TMC, GH-TPC, and Melem-TPC (GH: guanidine hydrochloride, TMC: acid chloride, BTCA: benzene-1, 3, 5-tricarbaldehyde, OC: oxalyl chloride, TPC: terephthaloyl chloride), are prepared via the polycondensation reactions between amine-containing precursors and carbonyl compounds (See synthetic details in "Methods"). Modulus analyses via atomic force microscopy (AFM) revealed that 2D polyamides with smaller rings exhibit higher Young's moduli (Fig. 1 and Supplementary Table 1). Specifically, the modulus of GH-TMC is determined to be 33.77 ± 4.06 GPa, which surpasses most 2D organic polymers. 2D polyimide of GH-BTCA with the almost identical ring unit to GH-TMC but lack of H-bons gives rise to a much lower modulus of 17.51 ± 1.36 GPa, revealing the role of H-bond in enhancing the modulus of 2D materials. The H-bond effect is also verified by the higher modulus of GH-TPC as comparison with Melem-TMC, because the former containing triple H-bonds in the structural ring unit.

## Results

### Structural characterization of 2D polyamide

Specifically, GH-TMC is prepared via an irreversible amide condensation reaction between C3-symmetric GH and TMC (Fig. 2a and Supplementary Fig. 1). GH molecules and TMC molecules react to form six-membered ring structural units with a triple H-bond between adjacent amide bonds. Mass spectrometry and solid-state $^{13}$C nuclear magnetic resonance (NMR) spectra (Supplementary Figs. 2, 3) confirms the presence of the six-membered ring structure in GH-TMC, consistent with the designed framework. X-ray photoelectron spectroscopy (XPS) of the C1s spectrum reveals a convolution peak at a binding energy of 292.1 eV (Supplementary Fig. 4), indicative of the presence of C$^+$ ions from guanidium. GH-TMC is supposed to adopt a staggered AB stacking configuration as guanidium ions bearing positive charges repel the aligned face-to-face arrangement (Fig. 2b). The crystallinity of the GH-TMC nanosheets was subsequently verified by powder X-ray diffraction (PXRD). The diffraction peak responding to (100) plane reflects the in-plane ordered arrangement, while the (001) peak indicates an interlayer spacing of 3.38 Å (Fig. 2c). Furthermore, the diffraction rings observed on the $q_z$ axis by wide-angle X-ray scattering

(WAXS) also manifest the interlayer spacing in the z direction. Peaks in the 1D profile are observed at approximately 1.27 Å$^{-1}$ and 1.86 Å$^{-1}$, respectively (Fig. 2d, e). The former corresponds to the periodic in-plane structure, agreeing with the previously simulated intertwined AB stacking mode; while the latter represents the π-π stacking distance between adjacent layers in 2D GH-TMC, with a spacing of 3.38 Å, aligning with the PXRD and simulation results (Fig. 2f). This interlaced AB stacking method does not cause the pores or rings within the molecules to be neatly arranged, consistent with the low Brunauer-Emmett-Teller (BET) surface area (Supplementary Fig. 5). The structural characterizations of other 2D polymer materials using Melem as a reactive unit are provided in Supplementary Figs. 6–8.

As the reaction of GH and TMC proceeds, the disappearance of the acyl chloride vibration peak at 1793 cm$^{-1}$ as represented by the black dashed line in the Fourier-transform infrared (FTIR) spectrum, confirmed the reaction's completion (Supplementary Fig. 9). Meanwhile, the vibrational peak in the highlighted yellow region verified the formation of amide bonds. The amide bond serves as a connecting node in GH-TMC, imparting robust H-bonding interactions between neighboring amide motifs (Fig. 2a, g). Evidence of these intermolecular H-bonds was confirmed by NMR spectroscopy (Supplementary Fig. 10). Single-pulse experiments carried out at different GH-TMC concentrations demonstrated a downfield chemical shift of the amide protons. This phenomenon indicates the participation of these protons in intermolecular H-bonding[39]. Additional support for these H-bond interactions among amide comes from Raman spectroscopy analysis (Supplementary Fig. 11). The intensity ratio (ρ) between peaks at 860 and 830 cm$^{-1}$ in the Raman spectrum is calculated to be 3.1, substantially greater than 1. This elevated ρ value points to notable hydrophilic interactions, further confirming the presence of strong intramolecular H-bonds[40] (Fig. 2g). Using an established infrared spectroscopy method[41], the average number of H-bonds per amide bond was estimated to be approximately 2.54 (Supplementary Fig. 12). This is close to high-performance aramid materials such as Kevlar (~2)[42], and significantly exceeds typical nylon (1.3 ~ 1.8)[43], highlighting the strength and prevalence of the H-bonding network within GH-TMC. Note that intramolecular H-bonds exist in the ring structure, while intermolecular H-bond interaction occurs at the limbic edge among adjacent nanosheets. There is no H-bond between contiguous monolayers as the experimental interlayer spacing of 3.38 Å is typical for the distance of π-π interaction, not suitable for external H-bonding (Fig. 2c). The resulting 2D amide polymer consists of extremely small structural units, endowing it with a remarkably high modulus. Furthermore, the strong H-bonding network formed by the amide linkages in the GH-TMC framework greatly enhances its modulus as indicated in Fig. 1.

### Fabrication of 2D polyamide films

AFM was used to characterize the dimensions of the 2D polymers (Supplementary Figs. 13–15). The nanosheets exhibit a pronounced aggregation tendency due to intralayer interaction, including π-π and malposed electrostatic interactions (Fig. 3a, b). Due to the presence of C$^+$ ions from guanidium in the GH-TMC structure, Cl$^-$ ions act as counterions to pull each GH-TMC oligomer through electrostatic interaction, thus forming a stable nanosheet structure. The presence of Cl$^-$ in the GH-TMC sample was proved by Cl 2p XPS peak (Supplementary Fig. 4d) and elemental mapping in scanning electron microscopy (SEM) images (Supplementary Fig. 16). Measurements of lateral dimensions and thickness of GH-TMC nanosheets reveal an average lateral size of 40.66 ± 10.41 nm (Fig. 3c) and an average thickness of 5.69 ± 3.99 nm (Fig. 3d), respectively. Further examination by high-resolution transmission electron microscopy (HR-TEM) showed the morphology of GH-TMC featured the stacked nanosheets, with dimensions consistent with those observed by AFM (Fig. 3e, f). At the sample edges, distinct layered structures were visible, a characteristic

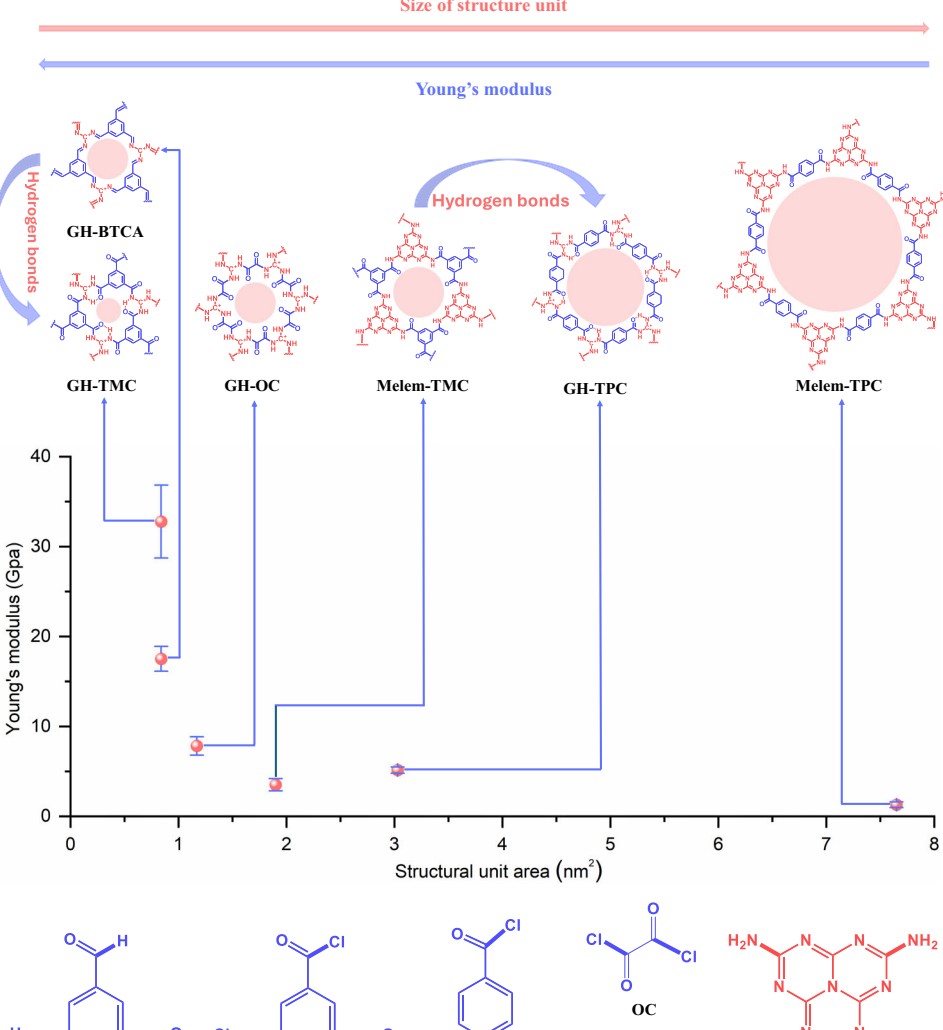

**Fig. 1 | Two-dimensional polymer materials with different structural units.** The figure illustrates a series of 2D polymer structural units alongside their corresponding Young's modulus values. It is evident that 2D polymer films with smaller structural units exhibit higher moduli, primarily due to the rigidity of these structural components. In addition, the presence of strong H-bonding networks significantly enhances the Young's modulus, thereby increasing the strength of the 2D polymer films. Note that the Young's modulus values presented in the figure were measured using the PF-QNM mode of AFM. Data are presented as mean values +/− SD, where the number of replicates $n = 3$. (GH guanidine hydrochloride, TMC acid chloride, BTCA benzene-1, 3, 5-tricarbaldehyde, OC oxalyl chloride, TPC terephthaloyl chloride).

that was consistently observed for 2D polyamide samples (Supplementary Figs. 17, 18). These results suggest that GH-TMC nanosheets can stack through misaligned H-bonds at the edges, creating an orderly planar stacking structure. This organization offers ideal conditions for self-assembly and facilitates film formation.

Building on these findings, we processed the 2D polymer material into films to explore its potential applications. Unlike other 2D COF materials, strong crystallinity and random stacking patterns inhibit their film formation[44,45]. GH-TMC nanosheets can be dispersed well in polar solvents such as ethanol owing to their small sheet size and abundant polar groups at the edges, which further make GH-TMC nanosheets liable for effective stacking through multiple weak interactions. The ethanol dispersion was first spin-coated onto a silicon wafer substrate with a smooth surface to achieve uniform coverage, followed by heating to evaporate the solvent for film fabrication. During its formation process, the structure formed by the dispersion of GH-TMC in ethanol solution was composed of some oligomer fragments. It was confirmed by mass spectrometry and $^{13}$C nuclear magnetic resonance spectroscopy that it was an oligomer composed of six-membered ring structural units formed by GH and TMC. Due to the presence of the counter anion Cl⁻, with the evaporation of the solvent, the GH-TMC oligomers formed AB stacked nanosheets through π-π and dislocation electrostatic interactions. The TEM and AFM images of these 2D GH-TMC nanosheets were shown in Fig. 3e, and their average transverse size was about 40 nm. These small-sized GH-TMC nanosheets can be stacked through edge-displaced H-bonds, thereby forming large nanosheet structures (Fig. 3f). These larger nanosheets further formed ordered planar stacked structures through electrostatic interactions and interlaced edge H-bonds, thereby forming complete films (Fig. 3g, h). SEM (Fig. 3i and Supplementary Fig. 19) and AFM analyses revealed that the film exhibited minimal surface roughness, with a root mean square roughness of about 500 ppm (Supplementary Fig. 20). The film thickness was monitored through SEM cross-sectional imaging, and Fig. 3j illustrates a typical

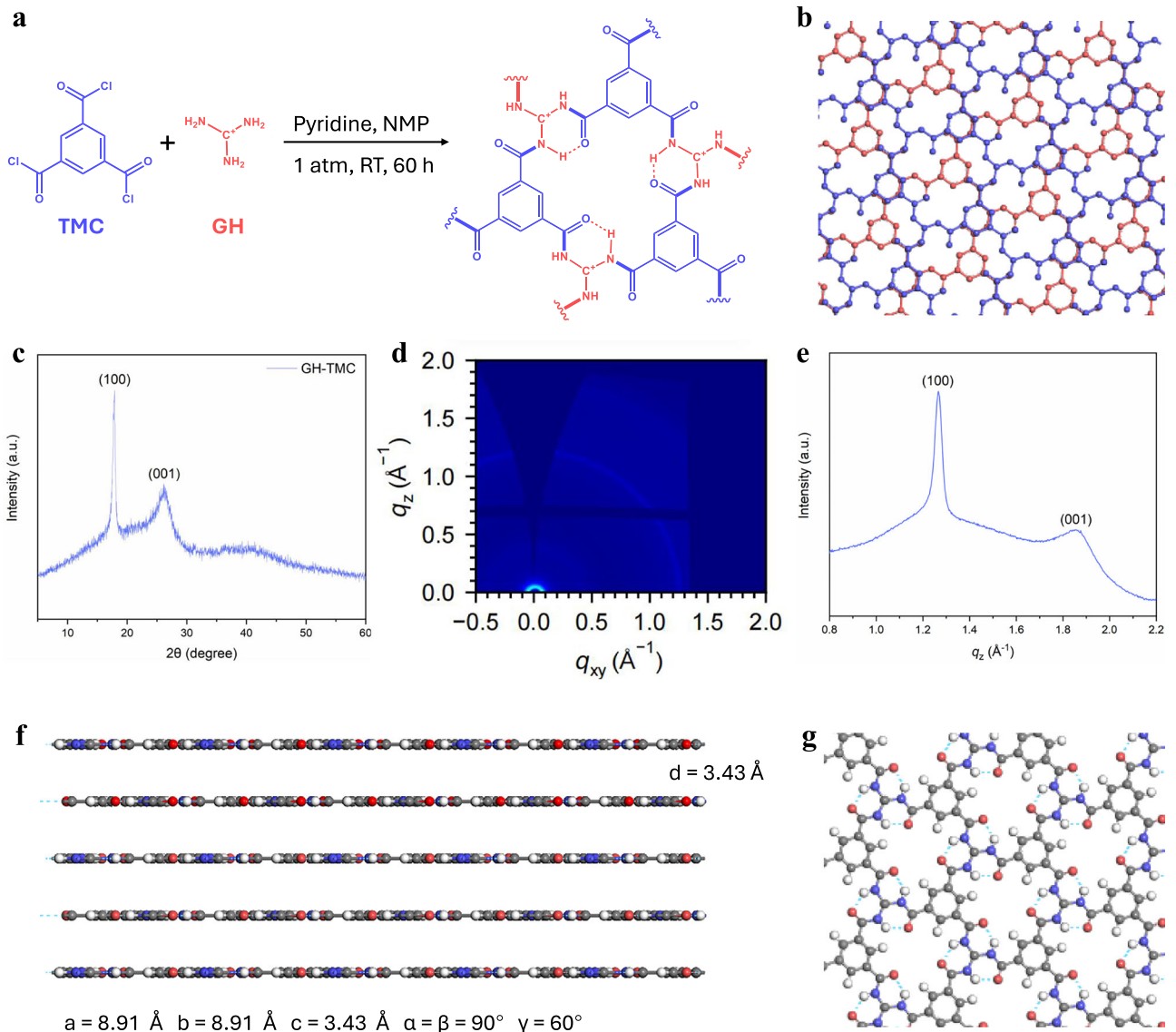

**Fig. 2 | Structure of GH-TMC. a** Synthetic route of 2D GH-TMC. **b** Simulated AB stacking structure of 2D GH-TMC. The hydrogen atoms are omitted for clarity, where blue and red are the adjacent upper and lower GH-TMC monolayers, respectively. **c** PXRD of GH-TMC. **d** WAXS 2D image of GH-TMC. **e** 1D intensity profile of GH-TMC. **f** Simulated AB stacking interlayer structure of GH-TMC film with an interlayer spacing of 3.43 Å. **g** Simulated H-bond (blue dashed line) in 2D monolayer GH-TMC.

film with a thickness of 800 nm, in accordance with AFM measurements (Fig. 3k). The film thickness could be fine-tuned by adjusting the concentration of the dispersion, allowing for a range from ultra-thin 20 nm to micrometer-scale thicknesses (Supplementary Figs. 21, 22). Notably, the prepared film can be transferred onto various substrate types, which further facilitates its applicability. Although some partial breakage may occur during transfer, continuity and flatness of the film retains across the substrate (Supplementary Fig. 23).

### Mechanical properties of the 2D GH-TMC film
Complete, continuous and flat films make it possible to study their mechanical properties. To investigate the mechanical properties, we measured the Young's modulus of a series of 2D polymer films using AFM in PF-QNM mode (peak force tapping mode quantitative nanomechanical imaging) (Supplementary Fig. 24). Note that the areas of the structural units were calculated as regular hexagons (Supplementary Fig. 25). The results as shown in Fig. 1 indicate that the GH-TMC film with the smallest structural unit exhibits the highest Young's modulus at 33.77 ± 4.06 GPa, significantly surpassing other 2D polymer

films. To further explore the role of H-bonding in enhancing mechanical strength, we synthesized GH-BTCA, a structurally similar polymer to GH-TMC, but with imine bonds as connecting nodes (Fig. 4a and Supplementary Fig. 26). This modification prevents GH-BTCA oligomer from participating in H-bonding interactions. The Young's modulus of GH-BTCA was markedly lower than that of GH-TMC, which bears triple H-bonds (Fig. 4b, c). This comparison clearly proves the strength enhancement effect of H-bonding interactions on 2D polymer films.

To enhance the modulus of 2D polymer materials, the incorporation of H-bond and careful design of structural units are effective strategies. The H-bond density in 2D polyaramid films plays a critical role in determining their mechanical strength and stability (Supplementary Note 1). To analysis the effect of the H-bonds within the 2D structure, we choose forcite module to do geometry optimization and dynamic molecular computations. To gain a better understanding of the H-bonds interaction, we firstly expanded the unit cell into a 5 * 5 * 2 supercell. The Molecule dynamics (MD) simulations were run for total 1000 ps to analysis the H-bonds (Supplementary Fig. 27 and

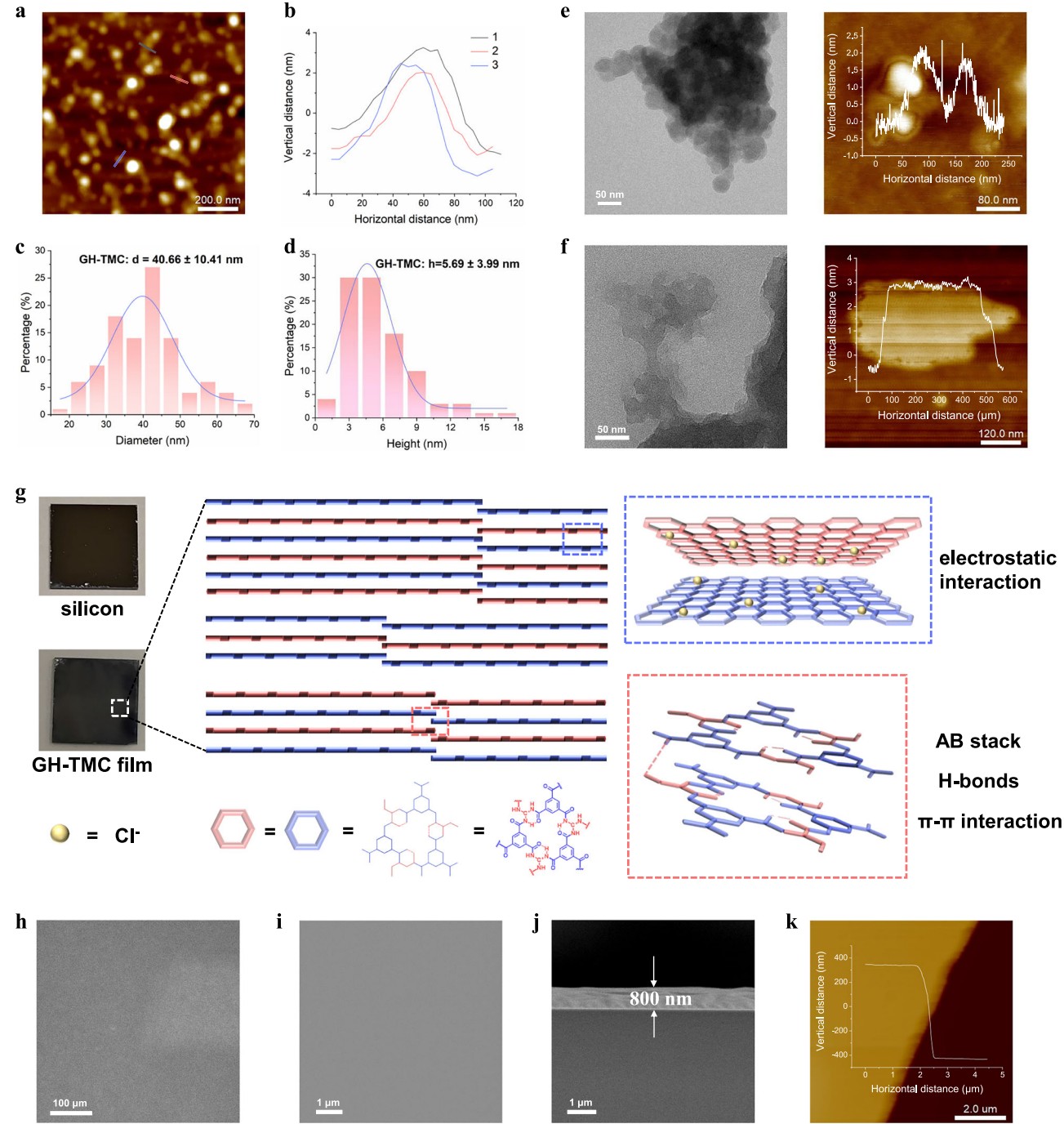

**Fig. 3 | Characterization of 2D GH-TMC nanosheets and film. a** AFM image of GH-TMC nanosheets. **b** Height profiles along the different colored lines indicated in (**a**) (from top to bottom: black, red and blue). **c**, **d** Size and height distribution of observed GH-TMC nanosheets in (**a**), respectively. **e** TEM image and high-resolution AFM image of GH-TMC nanosheets. Inset: Height profile of single GH-TMC nanosheet. **f** TEM image and high-resolution AFM image of stacked GH-TMC nanosheets. Inset: Height profile of single stacked GH-TMC nanosheet. **g** Digital

photograph of silicon substrate and GH-TMC film deposited on substrate and film forming diagram (1 cm * 1 cm), showing the malposed stacking structure, multiple interactions between adjacent monolayers and nanosheets. **h** Optical microscope image of GH-TMC film. **i**, **j** Top-view and cross-sectional SEM images of GH-TMC film. **k** AFM image of the morphology of the 2D GH-TMC film on monocrystalline silicon wafer. Scale bars, 200 nm (**a**), 50 nm (**e**) and (**f**), 100 μm (**h**), 1 μm (**i**) and (**j**), 2 μm (**k**).

Supplementary Movie 1). GH-TMC not only has multiple H-bonds within the plane, but also has interlaced H-bonds at the edges for connecting the nanosheets. In this supercell, the number of H-bonds simulated and calculated is between 340–364. Therefore, we can calculate that the number of H-bonds in the amide bond of GH-TMC is approximately 2.35, which is extremely close to the number of H-bonds we have tested through the established infrared spectroscopy method. The enhanced mechanical properties in 2D GH-TMC stem

from the synergistic molecular-level H-bond network. The H-bonds within the GH-TMC six-membered ring molecules mainly control the stiffness of the material, thereby increasing its Young's modulus. Each six-membered ring in GH-TMC has triple H-bonds between adjacent amide groups (N-H⋯O = C), forming a rigid cyclic structure (Fig. 2a). This can increase the bond density, thereby enhancing the stiffness of the covalent network in the plane. Cyclic constraints and C3-symmetry reduce the rotational entropy of the bond and further increase the

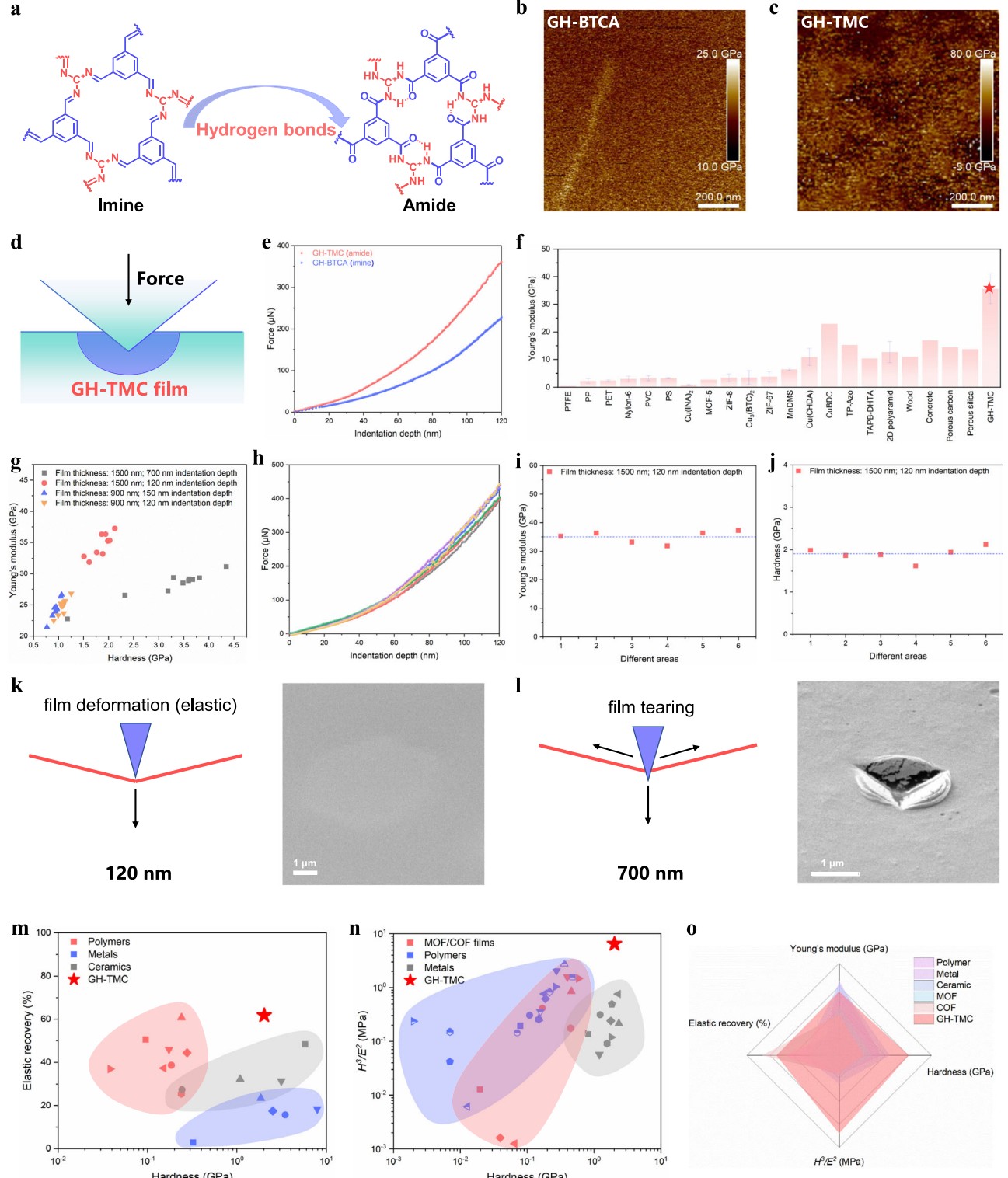

**Fig. 4 | Mechanical properties characterization of GH-TMC films. a** Structural units of GH-TMC (amide) and GH-BTCA (imine). **b**, **c** Young's modulus distribution of GH-BTCA and GH-TMC films, respectively. **d**, **e** The schematic diagram of the nanoindentation and corresponding force-displacement curves of GH-TMC and GH-BTCA films. **f** Comparison of Young's modulus of GH-TMC film with other typical materials. **g** Young's modulus versus hardness for different film thicknesses of GH-TMC and indentation depths. **h** Force-displacement curves of GH-TMC film in different regions. Note that the indentation depth is set at 120 nm. **i** Young's modulus from in situ SEM nanoindentation at different areas of the GH-TMC film. **j** Hardness from in situ SEM nanoindentation at different areas of GH-TMC film. **k**, **l** The schematic diagram of elastic and plastic deformation of GH-TMC film, respectively. **m** Elastic recovery rate and hardness ($H$) of GH-TMC film compared with those of typical polymers, metals and ceramics. **n** Comparison of $H^3/E^2$ versus $H$ for GH-TMC film and other typical materials, including MOF/COF films, polymers and metals. **o** Mechanical performance of the 2D polyamide (GH-TMC) and of typical polymer (PP), metal (Al), ceramic (calcite), MOF (CuBDC) and COF (TP-Azo) with the Young's modulus, hardness, $H^3/E^2$ and elastic recovery plotted as a radar map. Scale bars, 200 nm (**b**) and (**c**), 1 μm (**k**) and (**l**).

Young's modulus. For the intermolecular H-bonds at the edges of nanosheets, when π-π stacking dominates the interlayer cohesion, the 3.38 Å interlayer spacing excludes the H-bonds between the basal planes, confining the intermolecular H-bonds to the edge regions. H-bonds form at the edges of the nanosheets and are confined to the junctions of different nanosheets, mainly controlling the elasticity of the material. The reversible H-bond fracture/reconstruction at the edge ensures a relatively high elastic recovery rate even under indentation. Among them, the energy dissipation from H-bond fracture alleviates crack propagation and helps to enhance toughness. This double H-bond mechanism, by combining the rigidity of covalent networks and multiple weak interaction forces, endows GH-TMC films with excellent mechanical properties.

In order to ensure the accuracy of our Young's modulus measurements, we also analyzed the mechanical properties using in situ SEM nanoindentation. The schematic diagram of the nanoindentation is shown in Fig. 4d. The technique not only provided stress-displacement curves but also enabled real-time observation of indentation and potential film damage. To minimize substrate influence on the film indentation test, we prepared all films with micrometer-level thickness and kept the indentation depth at 120 nm, less than 10% of the membrane thickness so that to ensure the indentation occurring in the elastic range. In situ SEM indentation (using a triangular pyramid-shaped Berkovich indenter) was tested on a randomly selected 2D film with a fixed indentation depth of 120 nm. Fitting the stress-displacement curves revealed a nonlinear elastic response, and the Young's modulus values for GH-TMC and GH-BTCA films were determined to be 35.683 and 13.862 GPa, respectively (Fig. 4e). These results closely aligned with AFM measurements, further confirming the reliability of our findings.

Due to the presence of multiple interactions, including π-π, malposed electrostatic H-bond interaction in the film as well as the rigid structural units, GH-TMC exhibits seemingly contradictory mechanical properties: it has the strength and hardness of metallic materials, and the resilience of polymer materials. According to the load displacement curve, the Young's modulus and hardness $H$ of GH-TMC film are $35.612 \pm 5.394$ GPa and $2.021 \pm 0.615$ GPa, respectively (Supplementary Fig. 28). Its Young's modulus and hardness exceed that of almost all polymer materials, COF/MOF materials, and even some metal materials (Fig. 4f and Supplementary Table 2). For example, traditional 1D polymers, such as nylon and polycarbonate, typically exhibit a modulus of 2–4 GPa[46–49], and semi-crystalline polyethylene terephthalate (PET) also falls within this range[50]. Among 2D polymers, the modulus of GH-TMC far overwhelms typical 2D COFs (10.38 GPa)[8] and 2D MOFs ($10.9 \pm 3.1$ GPa)[51].

Further, we performed indentation tests at different depths on films with different thicknesses (Fig. 4g). The elastic recovery rate can be used to evaluate the resilience of the film[52]. When the indentation depth is 120 nm, the stress-displacement curve remains consistent under the same test conditions (Fig. 4h). The Young's modulus and hardness measured at different positions of the film are basically the same, which indicates that the film has overall uniformity and strong stability (Fig. 4i, j). No obvious residual impression after indentation was observed in SEM images, indicating that the GH-TMC film has good resilience (Supplementary Fig. 29 and Supplementary Movie 2). Due to the stacked GH-TMC nanosheets are connected to each other by H-bonds at the edges, a lamellar interlock structure is formed. This enables the GH-TMC membrane to maintain its resilience under the action of a certain external force (Fig. 4k). Even if we apply a much larger force, it causes the indentation depth to reach 700 nm. After six consecutive indentations, the stress-displacement curve of GH-TMC film remains unchanged, which shows that the film has remarkable toughness and stability (Supplementary Fig. 30). The SEM image of the damaged area of the film is shown in Fig. 4l. Impressively, permanent plastic deformation occurs on the surface of GH-TMC film, the elastic recovery rate is still as high as 60% (Supplementary Figs. 31, 32). This is comparable to polymer materials and much higher than metal materials and some ceramic materials, indicating that GH-TMC film have elasticities similar to rubber (Fig. 4m and Supplementary Table 3).

In addition, the $H^3/E^2$ value is related to the yield pressure of different contact modes and is used to evaluate material resistance to initiation of abrasive damage[53,54]. The $H^3/E^2$ value of GH-TMC film is higher than that of polymer and metal materials, by comparing the $H^3/E^2$ value and hardness value with other materials, indicating its strong wear resistance (Fig. 4n and Supplementary Tables 4–6). GH-TMC film exhibits superior overall mechanical properties in terms of Young's modulus, elasticity, hardness and yield strength, as compared to other typical materials (Fig. 4o), positioning it as a promising candidate for high-performance applications.

## Discussion

For most 2D materials or polymeric materials, strength and resilience are often considered contradictory properties, with few materials effectively balancing both. In this work, we synthesized GH-TMC films that achieve a remarkable combination of strength and resilience by simultaneously reducing the size of rigid structural units, increasing the chemical bond density, incorporating a robust H-bond network, with π-π and electrostatic interactions as well. The smaller rigid structural units contribute to enhanced strength, while the interwoven H-bond network not only reinforces the film's strength but also imparts high resilience, resulting in exceptional toughness. The impressive modulus and exceptional resilience highlight the superior mechanical properties of GH-TMC, establishing it as a promising candidate for high-performance applications. This work points out a general strategy for fabricate 2D metamaterials by incorporating the multiple interactions into the system, which could further guide the 2D material design for specific applications.

## Methods

### Materials

All materials were purchased commercially and used without further purification. Guanidine hydrochloride ($CH_6ClN_3$, 99%), melamine ($C_3H_6N_6$, 99%), benzene-1, 3, 5-tricarbaldehyde ($C_9H_6O_3$, 96%) and N-methyl-2-pyrrolidone (98%) were purchased from Shanghai Macklin Biochemical Co., Ltd. (China). Terephthaloyl chloride ($C_8H_4Cl_2O_2$, 99%), 1, 3, 5-benzenetricarboxylic acid chloride ($C_9H_3Cl_3O_3$, 98%), oxalyl chloride ($C_2Cl_2O_2$, 98%), pyridine ($C_5H_5N$, 99.5%) and tri-fluoromethanesulfonic acid ($CF_3SO_3H$, 99%) were purchased from Adamas-beta Co., Ltd. (China). Oxolane ($C_4H_8O$, 99%), ethyl alcohol ($C_2H_6O$, 99.7%) and acetone ($C_3H_6O$, 99%) were purchased from Sinopharm Chemical Reagent Co., Ltd. (China).

### Syntheses of 2D polymeric materials

For the synthesis of two-dimensional polymers, we directly synthesized two-dimensional nanosheets through the self-catalysis and restricted rotation method reported previously[13], and then formed films by spin coating and heating to evaporate the solvent. Specifically:

**GH-TMC.** N-methyl-2-pyrrolidone of 9 mL was added as the solvent into a 50 mL glass vial, followed by guanidine hydrochloride (96 mg, 1 mmol) and 1,3,5-benzenetricarboxylic acid chloride (265 mg, 1 mmol). The mixture was stirred until fully dissolved. Subsequently, 1 mL of pyridine was added as a catalyst to initiate the reaction, which occurred rapidly, leading to gel formation. The mixture was stirred at room temperature for 60 h. After the reaction was complete, the mixture was washed sequentially with tetrahydrofuran (THF) and water. The resulting brownish-yellow powder was freeze-dried.

**GH-BTCA.** 9 mL of N-methyl-2-pyrrolidone was added as the solvent into a 50 mL glass vial, followed by the addition of guanidine

hydrochloride (96 mg, 1 mmol) and benzene-1,3,5-tricarbaldehyde (162 mg, 1 mmol). The mixture was stirred until fully dissolved. Subsequently, 1 mL of pyridine was added as a catalyst to initiate the reaction, and the mixture was stirred at 80 °C in an oil bath for 60 h. After the reaction was complete, the mixture was washed sequentially with tetrahydrofuran (THF) and water. The resulting yellow powder was freeze-dried.

**GH-TPC.** 9 mL of *N*-methyl-2-pyrrolidone was added as the solvent into a 50 mL glass vial, followed by the addition of guanidine hydrochloride (96 mg, 1 mmol) and terephthaloyl chloride (304 mg, 1.5 mmol). The mixture was stirred until fully dissolved. Subsequently, 1 mL of pyridine was added as a catalyst to initiate the reaction, and the mixture was stirred at room temperature for 60 h. After the reaction was complete, the mixture was washed sequentially with ethanol, acetone, and water. The resulting brown powder was vacuum-dried at 80 °C for 12 h.

**GH-OC.** 9 mL of *N*-methyl-2-pyrrolidone was added as the solvent into a 50 mL glass vial, followed by the addition of guanidine hydrochloride (96 mg, 1 mmol) and 1 mL of pyridine. The mixture was stirred until fully dissolved. It is important to note that, due to the rapid reaction, 200 μL of oxalyl chloride must be added dropwise while stirring the mixture at a low temperature. After the reaction was complete, the mixture was washed sequentially with tetrahydrofuran (THF) and water. The resulting brown powder was freeze-dried.

**Melem-TMC.** First, Melem powder was synthesized according to a previously reported method[55]. Melamine (1 g) was heated to 450 °C under an air atmosphere and held for 5 h, with a heating rate of ~1 °C min⁻¹ and a cooling rate of about 2 °C min⁻¹, yielding Melem powder. 100 mg of Melem powder and 500 mg of 1,3,5-benzenetricarboxylic acid chloride were ball-milled and mixed uniformly. The mixture was then heated to 350 °C under an air atmosphere and held for 5 h, with a heating rate of approximately 3 °C min⁻¹ and a cooling rate of about 2 °C min⁻¹. The resulting product was washed sequentially with ethanol, trifluoroacetic acid and water. The brown powder obtained was vacuum-dried at 80 °C for 12 h.

**Melem-TPC.** 100 mg of Melem powder and 500 mg of terephthaloyl chloride were ball-milled and mixed uniformly. The mixture was then heated to 350 °C under an air atmosphere and held for 5 h, with a heating rate of approximately 3 °C min⁻¹ and a cooling rate of about 2 °C min⁻¹. The resulting product was washed sequentially with ethanol, trifluoroacetic acid and water. The yellow powder obtained was vacuum-dried at 80 °C for 12 h.

### Preparation of 2D polymeric films
Typically, 2D polymeric materials were dispersed in ethanol, then spin-coated on a single-crystal silicon wafer. Finally, the solvent is evaporated by heating to form a film. The thickness of films was controlled by adjusting the concentration of the dispersion. Note that, Melem-TMC and Melem-TPC samples were dispersed in trifluoromethanesulfonic acid (TfOH) solution.

### Powder X-ray diffraction
Powder X-ray diffraction (XRD) measurement was carried out on a Rigaku MiniFlex 600 X-ray diffractometer using Cu Kα radiation ($\lambda = 1.54178$ Å). All diffraction patterns were collected between 2θ angles of 3° to 60°.

### Wide-angle X-ray scattering
The crystal orientation of 2D GH-TMC was determined by the WAXS method. The measurement for 2D GH-TMC was on the X-ray Scattering System with the instrument model SAXS point 2.0. The distance from the sample to the detector was 113 mm.

### Nuclear magnetic resonance spectroscopy
The Solid state ¹³C NMR spectra of the GH-TMC experiment were performed using a 14.1 T Bruker AVANCE NEO 600 WB spectrometer equipped with a 4 mm low γ probe with a sample size of 100 μL. Segments of single-pulse spectra of GH-TMC at different concentrations were obtained using a 600 MHz superconducting Fourier nuclear magnetic resonance spectrometer operating at 14.1 T, and 32 scans were collected.

### Transmission electron microscopy
Transmission electron microscopy (TEM) experiments and selected electron diffraction were performed on a JEM-2100 Plus corrected at 200 kV. Data acquisition was carried out on an SIS QUEMESA 11 million pixels bottom-plug-in CCD camera.

### Atomic force microscope
Atomic force microscope (AFM) images were obtained on a Bruker dimension fast scan instrument (peak force mode). AFM nanoindentation was carried out with AFM in PF-QNM mode (peak force tapping mode quantitative nanomechanical imaging). And the force provided was maintained at 50 nN. All collected data are analyzed by the Nanoscope v9.2 software.

### In situ field emission scanning electron microscope nanoindentation
In situ SEM nanoindentation microstructure evolution images were obtained by SEM5000X. It is equipped with the Oxford FT-NMT04 in situ nanoindentation system, which can perform nanoindentation test on micro-nano scale materials in SEM and observe the film morphology in situ combined with SEM. In situ SEM indentation (using a triangular pyramid-shaped Berkovich indenter) was tested on a randomly selected 2D film with a fixed indentation depth of 120 nm. All collected data was analyzed using the Femto Tools Suite software.

### Other characterization methods
X-ray photoelectron spectrometer (XPS) measurements were carried out on the Thermo Scientific ESCALAB 250Xi spectrometer using a monochromatic Al kα 150 W source with a beam spot of 500 μm. Fourier transform infrared (FT-IR) spectroscopy of samples were tested on a Nicolet 6700 spectrometer with KBr discs. Raman spectra was obtained at LabRAM HR Evolution with an excitation wavelength of 1064 nm. The mass spectrum data was obtained by a stroma-assisted laser desorption tandem time-of-flight mass spectrometer. Scanning electron microscopy (SEM) images were carried out with a field-emission scanning electron microanalyzer (Gemini SEM 500).

### Details of simulation
The software we used for DFT calculation was version V6.1 of Materials Studio. The basis set was Plane-wave unit. Geometry optimization were performed with the CASTEP program within the framework of DFT[56,57]. The ultra-soft pseudopotential was used for electron-ion interactions, and the Perdew-Burke-Ernzerhof (PBE) form of the generalized gradient approximation (GGA) was employed to describe the exchange correlation functional[58]. The plane-wave cutoff energy was set at 600 eV, the sizes of the k-point meshes for Brillouin zone sampling of primitive cells were 3 * 3 * 6. The tolerance for the self-consistent field, maximal force, maximum displacement and maximum stress were set at $1.0 \times 10^{-6}$ eV/atom, $1.0 \times 10^{-2}$ eV/A, $1.0 \times 10^{-3}$ A, $2.0 \times 10^{-2}$ GPa.

To analysis the effect of the H-bonds within the 2D structure, we choose forcite module to do geometry optimization and dynamic molecular computations. To gain a better understanding of the H-bonds interaction, we firstly expanded the unit cell into a 5 * 5 * 2 supercell. The relaxation of all the atoms was performed with the convergence tolerance was set to $2.00 * 10^{-5}$ and

0.001 kcal mol$^{-1}$ for maximum force. Van der Waals interactions were calculated using atom-based summation, and electrostatic interactions were treated using Ewald summation with a cutoff distance of 18.5 Å and a buffer width of 0.5 Å. COMPASS II force field was applied here as it is suitable for our system. Molecule dynamics (MD) were carried out after geometry optimization in NPT ensembles (N: number of particles; P: pressure; T: temperature) at 0.1 MPa and 298.15 K by using the Nosé thermostat. The MD simulations were run for total 1000 ps to analysis the H-bonds.

## Data availability
All data generated or analyzed during this study are included in this article and its Supplementary Information files, other data that support the findings of this study are available from the corresponding author upon request.

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

## Acknowledgements

We acknowledge support from the Chinese Academy of Sciences and the University of Science and Technology of China, National Key Research and Development Program of China (2021YFA1500400), National Natural Science Foundation of China (NSFC, 22471252, 21571167, 51502282 and 22075266), Fundamental Research Funds for the Central Universities (WK2060190053 and WK2060190100).

## Author contributions

B.L. came up with the idea and designed these experiments. Q.H. synthesized samples, prepared films, and performed all tests. CR.W. helps with structural simulation and calculation. Y.Z. and AW. H. helped with the characterization of the samples and films.

## Competing interests

The authors declare no competing interests.
