## [Transparent Peer Review file · Nature Communications]

Manipulating mechanical strength of isorecticular two-dimensional polyamide materials via multiple interactions

Corresponding Author: Professor Bo Liu

Version 0:

Reviewer comments:

Reviewer #1

(Remarks to the Author)

Liu et al. present a facile synthesis approach for a novel two-dimensional polymer with an unusual structure that can be readily dispersed and reassembled into robust polymer films. The structural design is intriguing, and the experimental results are generally convincing. However, the explanation of the mechanical properties remains insufficiently supported. The following questions should be addressed prior to publication in Nature Communications:

1. The AFM images indicate that the 2DP particle size is very small (on the order of tens of nanometers), and the GIWAXS results suggest that the particles in the films are randomly oriented. Please provide a more detailed explanation of how such ultra-small, randomly oriented particles give rise to the observed strong mechanical properties of the stacked film. For instance, is there any evidence of edge-to-edge bonding or other forms of interparticle connectivity that might contribute to film integrity? The current explanation, which attributes the strength to "multiple interactions," is not sufficiently convincing to account for the unusual mechanical robustness and high elastic recovery observed considering the small particle size.
2. The GIWAXS peaks of both the dispersed particles and the assembled film are notably sharp, suggesting a high degree of crystallinity, but the particle size is rather small. Could the authors estimate the approximate crystallite size based on the GIWAXS data (e.g., using the Scherrer equation or similar analysis)?
3. The authors assign the peak between 1.8 and 1.9 Å⁻¹ as the (002) reflection; however, this peak is more likely to correspond to π-π stacking and should be indexed as the (001). Please double-check this assignment. Additionally, performing an XRD simulation based on the proposed structure would be helpful to validate the peak indexing.
4. Including BET analysis to determine the specific surface area and pore size distribution of the particles would be helpful in validating the proposed porous structure.
5. What is the upper thickness limit for the assembled film? If the film can be fabricated with sufficient thickness, is it possible to produce a macroscopic, freestanding version?
6. The authors claim that multiple interactions contribute to the mechanical strength of the film. Could the authors clarify the relative contribution of each type of interaction?
7. For a more comprehensive mechanical comparison, the authors should consider including relevant literature on MOF/COF films/crystals and hybrid structures:
<https://www.pnas.org/doi/abs/10.1073/pnas.2208676120>
<https://www.science.org/doi/abs/10.1126/science.ads4968>
<https://www.science.org/doi/full/10.1126/science.adf2573>
<https://www.science.org/doi/10.1126/science.aad4011>
<https://www.nature.com/articles/s41467-020-15281-1>
<https://www.nature.com/articles/s41467-024-53935-6>

Reviewer #2

(Remarks to the Author)

This manuscript reports the synthesis and mechanical characterization of a series of isorecticular two-dimensional (2D) polyamide materials, emphasizing the influence of structural unit size and multiple non-covalent interactions (hydrogen bonding, π-π stacking, electrostatics) on their macroscopic mechanical performance. The work centers on the GH-TMC system, which exhibits an exceptionally high Young's modulus (up to ~35 GPa) and elastic recovery, outperforming many

state-of-the-art 2D polymeric materials and even some metallic films. While the study is of high potential interest and well supported by experimental data, there are a number of important scientific and presentation issues that need to be addressed before the manuscript is suitable for publication. I therefore recommend major revision, and outline specific comments below.

Abstract:

1. Please define all abbreviations at their first occurrence, including GH and TMC. Also ensure consistency throughout the manuscript (e.g., abbreviation undefined in line 53).
2. The abstract focuses exclusively on the mechanical performance of the materials. Please also highlight the broader significance of the work—for example, potential applications of these robust 2D polyamide films.

Main text:

1. Lines 48–51: The reason why bulk 2D materials are difficult to prepare at macroscopic scale is not clearly explained. Please revise this section for clarity and better logical flow.
2. The authors should more explicitly highlight the importance of enhancing mechanical strength in 2D polymers and discuss potential application fields where such materials are in demand. This will strengthen the motivation and innovation of the work.
3. In Figure 1, it would be helpful to include chemical structures of all monomers (e.g., GH, TMC, OC, etc.) to improve clarity.
4. Lines 66–70: The authors claim that GH and TMC form six-membered rings, but the provided mass spectrometry and NMR data (Supplementary Figs. 2 and 3) do not conclusively support this. Moreover, based on Figure 1 and 2a, the repeating unit seems more triangular in shape rather than hexagonal. This raises concerns about the validity of the structural model used for calculating the Young's modulus in Supplementary Fig. 24. The use of a regular hexagon approximation appears questionable.
5. Figure 2f shows a perfectly flat stacking model, which seems unrealistic given the nature of the oligomers. Have the authors performed any energy minimization or geometry optimization? Likewise, in Figure 2g, the hydrogen bonding pattern appears to be intramolecular rather than between nanosheets—please clarify this point. Furthermore, the comparison between H-bond density in GH-TMC and water is not appropriate. A more meaningful comparison would be with ordinary amide systems lacking the GH-based structure.
6. Please provide a more detailed molecular-level explanation of how hydrogen bonding contributes to the mechanical enhancement of 2D materials. The current discussion is somewhat descriptive and would benefit from deeper mechanistic insight.
7. Apart from the GH-TMC/TPC/OC and Melem-TMC/TPC materials reported, are there data available for a Melem-OC sample? A comparison between GH-based and Melem-based materials (e.g., GH-TMC vs. Melem-TMC) would help illustrate how monomer structure affects film properties.
8. Lines 172–182: The authors discuss the role of H-bond density in strengthening the films, but the argument is quite superficial. Can this be supported by molecular dynamics (MD) or density functional theory (DFT) simulations to provide quantitative insights?
9. Besides monomer structure, are there other factors (e.g., stacking configuration, counterion effects, defects) that influence the mechanical properties? These aspects deserve further discussion.
10. The manuscript suggests that smaller structural units (or possibly smaller pore sizes) lead to higher modulus values. Please elaborate on the physical rationale behind this trend.
11. The manuscript heavily emphasizes mechanical properties but does not discuss possible applications. For example, if these materials were to be used as membranes for separation, how would their high strength affect permeability? What are the typical pore sizes? Is there a trade-off between strength and permeability?
12. The authors also showed the film fabrication based on spin-coating from solution. Please comment on: (1) the reproducibility and scalability of this method; (2) whether these films can be produced over large areas with uniform thickness and consistent quality; (3) how do these films behave on flexible or deformable substrates?

Language and Presentation:

1. The manuscript contains numerous grammatical and typographical issues. For example, “As the reaction of GH and TMC proceeding...” should be revised to “As the reaction of GH and TMC proceeds...”. Also, use of “etc.” should be correctly and carefully used.
2. Please ensure all abbreviations are defined upon first use and avoid repeated definitions (e.g., AFM is defined in both line 56 and line 114).
3. A thorough language edit by a native English speaker or a professional editing service is strongly recommended to improve the clarity, consistency, and readability of the manuscript.

Reviewer #3

(Remarks to the Author)

This work presents an interesting study by achieving an impressive modulus of 35.6 GPa along with a 60% elastic recovery in the GH-TMC material, through precise molecular design and the synergistic contribution of multiple interactions, including hydrogen bonding, π - π stacking, and electrostatic forces. This effectively overcomes the longstanding trade-off between strength and elasticity in conventional materials. However, for improved clarity and rigor, the authors are encouraged to consider and elaborate on the following concern/points:

The main conclusion of this manuscript lies in the outstanding mechanical properties of the measured thin films. However, the current methodology relies on nanoindentation measurements performed on substrate-supported thick films, which inherently introduces potential substrate effects that cannot be completely excluded. To strengthen the validity of the reported mechanical parameters, the authors are advised to conduct additional mechanical tests on suspended/freestanding thin films—such as indentation or tensile experiments of GH-TMC. Comparative analysis between the results from suspended and substrate-supported configurations would enable cross-validation of the data/conclusions and provide more robust evidence for the intrinsic mechanical behavior of the films.

The overall writing and figure presentation in the manuscript require significant improvement. Specifically, Figures 3 and 4 in the main text are not sufficiently clear and need to be revised for better readability and visual quality. In the second paragraph of the Introduction, the authors list several materials such as 2D polymers, COFs, Kevlar, and graphene. However, this section lacks a synthesis of the common challenges or features among these materials. It is recommended that the authors highlight the theme of “multiple interactions” in the title by discussing the role of different types of chemical bonding/interactions shared by these materials.

Furthermore, to improve the clarity and logical structure of the manuscript, it is suggested that the authors organize the main text into sections that separately discuss how each type of interaction contributes to the mechanical enhancement of the material. This would provide a more systematic and coherent framework for presenting the results.

Additional technical comments

The meaning of the blue arrow in Figure 1 is unclear. If the authors intend to illustrate the relationship between mechanical strength and structural units, it is necessary to provide quantitative data to support this trend. Alternatively, the authors may consider using a dual-axis plot to more clearly convey the correlation between strength and structural features.

In Figure 2, the authors need to clearly distinguish between simulation results and experimental data, particularly in panels 2c–2e. It is important to explicitly label or annotate the corresponding data sources to avoid confusion and to ensure accurate interpretation of the results.

In Figure 3i, the thin film material is not clearly visible. If the authors intend to emphasize the flatness or uniformity of the film, it is recommended to include roughness analysis data based on AFM characterization in the main text to support this point more convincingly.

In Figures 4m and 4n, the authors use identical symbols to represent different materials, which makes it difficult to interpret the comparison. A unified and consistent comparison scheme is necessary. Moreover, the categories “ceramics” and “metals” encompass a wide range of materials; the authors should clarify which specific types of ceramics and metals are being compared. It is also strongly recommended to include a comparison with representative two-dimensional materials, such as graphene, to provide a more meaningful benchmark.

Version 1:

Reviewer comments:

Reviewer #1

(Remarks to the Author)

The authors have addressed most of my concerns, and I believe the manuscript is suitable for publication in Nature Communications in its current form.

Reviewer #2

(Remarks to the Author)

1. In the response, the authors reported that the average number of hydrogen bonds per amide unit in GH-TMC was ~2.35 (based on MD and IR analysis). However, in bulk liquid water, the average number of hydrogen bonds per water molecule is typically 3~4 according to the literature (DOI: 10.1038/ncomms9998). This raises concern about the appropriateness of comparing the hydrogen bonding environment of your 2D polymer network with that of bulk water. Since water is a small, highly dynamic hydrogen-bonding liquid, while your system is a rigid covalent network with constrained intramolecular H-bonds, the analogy may be misleading. A more relevant comparison would be with conventional polyamide systems (e.g., nylon, aramid), where intramolecular and intermolecular H-bonds play a well-recognized role in mechanical performance. I recommend revising this section to avoid comparison with water, and instead benchmark against structurally and chemically related polyamide networks.

2. The rebuttal emphasizes that most H-bonds in GH-TMC are intramolecular within the six-membered ring, with only marginal edge H-bonds between nanosheets. Since mechanical robustness is attributed to both stiffness and recoverability, could the authors more clearly separate the roles of intramolecular H-bonds (rigidity) and edge-localized intermolecular H-bonds (elastic recovery)? This distinction would improve mechanistic clarity.

3. The manuscript frequently refers to “multiple interactions” (π - π , electrostatic, H-bonds) as a synergistic source of mechanical strength. However, the relative contribution of each interaction type remains qualitative. While I appreciate the difficulty in full quantification, could the authors provide even an approximate estimation (e.g., through comparative calculations, or literature benchmarks) to show which interactions dominate under mechanical loading?

Reviewer #3

(Remarks to the Author)

The authors have addressed most of my review comments by conducting additional works. Although they cannot perform the free-standing stretching test due to the technical difficulty, considering the film thickness, the data from the indentation results can be acceptable.

Bo Liu, Ph. D., Professor
Department of Chemistry
University of Science & Technology of China (USTC)
96 Jinzhai Road, Hefei, Anhui 230026, P.R. China
Tel/ Fax: 86-551-63601123
Email: liuchem@ustc.edu.cn

Point-by-point responses to the reviewers' comments

(Reviewers' comments and the response are displayed in black and blue, respectively)

Reviewer #1 (Remarks to the Author):

Liu et al. present a facile synthesis approach for a novel two-dimensional polymer with an unusual structure that can be readily dispersed and reassembled into robust polymer films. The structural design is intriguing, and the experimental results are generally convincing. However, the explanation of the mechanical properties remains insufficiently supported. The following questions should be addressed prior to publication in Nature Communications:

Response: Thanks a lot for your time and effort on our manuscript.

Minor points:

1. The AFM images indicate that the 2D particle size is very small (on the order of tens of nanometers), and the GIWAXS results suggest that the particles in the films are randomly oriented. Please provide a more detailed explanation of how such ultra-small, randomly oriented particles give rise to the observed strong mechanical properties of the stacked film. For instance, is there any evidence of edge-to-edge bonding or other forms of interparticle connectivity that might contribute to film integrity? The current explanation, which attributes the strength to "multiple interactions," is not sufficiently convincing to account for the unusual mechanical robustness and high elastic recovery observed considering the small particle size.

Response: Thanks for your insightful comments. Taking the formation process of GH-TMC film as example, GH-TMC is firstly dispersed in ethanol solution, and oligomer fragments of GH-TMC composed of six-membered ring structural units formed by GH and TMC, was confirmed by mass spectrometry and ^{13}C nuclear magnetic resonance spectroscopy. Due to the presence of the counter anion Cl^- , with the evaporation of the solvent, the GH-TMC oligomers form AB stacked nanosheets through π - π and dislocation electrostatic interactions (characterized by XRD, GIWAXS and XPS). The TEM and AFM images of these 2D GH-TMC nanosheets are shown in Figure 3e, and their average transverse size is about 40 nm. These small-sized GH-TMC nanosheets can be stacked through edge-displaced hydrogen bonds (H-bonds), thereby forming large nanosheet structures (Figure 3f). These larger

nanosheets further form ordered planar stacked structures through electrostatic interactions and interlaced edge H-bonds, thereby forming complete films.

The high Young's modulus of GH-TMC films mainly comes from two parts. One is its extremely small ring unit structure. The smaller the ring unit structure, the larger the Young's modulus of the film tends to be (Figure 1). The nanosheets formed by the extremely small ring unit structure of GH-TMC are composed of AB stacking forms formed by π - π and offset electrostatic interactions. Smaller ring units (for example, the six-membered ring structure in GH-TMC) more effectively constrain molecular motion than larger rings, minimizing bond rotation and deformation under stress. This leads to a higher in-plane stiffness. Smaller units contain more chemical bonds per unit area while enhancing the covalent network's resistance to strain. Smaller units facilitate closer layer spacing and enhance the cohesion of the base plane through π - π interactions. Larger units may weaken this cohesion due to increased porosity or misalignment. Therefore, smaller structural units enhance mechanical strength by maximizing the density of covalent and H-bonds, while optimizing secondary interactions (π - π interactions, electrostatic interactions).

Besides, the in-plane H-bonds within the GH-TMC ring also enhance its longitudinal Young's modulus (Figure 4a-e), while the interlaced H-bond interactions at the edges enable the nanosheets to stack in an ordered plane, thereby forming a complete film, which also endows the film with a certain degree of resilience when subjected to longitudinal pressure. The H-bond interactions in GH-TMC have been confirmed and expounded in this article through characterizations such as FTIR, NMR and Raman spectroscopy. Each six-membered ring in GH-TMC has triple H-bonds between adjacent amide groups (N-H \cdots O=C), forming a rigid cyclic structure (Figure 2a), which increase the bond density, thereby enhancing the stiffness of the covalent network in the plane. Cyclic constraints and C_3 -symmetry reduce the rotational entropy of the bond and further increase the Young's modulus. For the intermolecular H-bonds at the edges of nanosheets, when π - π stacking dominates the interlayer cohesion, the 3.38 Å interlayer spacing excludes the H-bonds between the basal planes, confining the intermolecular H-bonds to the edge regions. H-bonds form at the edges of the nanosheets. The reversible H-bond fracture/reconstruction at the edge enables the elastic recovery under the indentation to reach 60%. Among them, the energy dissipation from H-bond fracture alleviates crack propagation and helps to enhance toughness. This double H-bond mechanism, by combining the rigidity of covalent networks and multiple weak interaction forces, endows GH-TMC films with excellent mechanical properties. Therefore, we believe that the high modulus and high elasticity of GH-TMC films are the result of multiple interactions such as their extremely small structural units, π - π and offset electrostatic interactions forming AB packing forms, and H-bond interactions in the formed films.

The specific influence of multiple interactions on the mechanical properties of GH-TMC films has been supplemented and discussed in detail in the Supplementary Note 1 of the Supplementary Information.

2. The GIWAXS peaks of both the dispersed particles and the assembled film are notably sharp, suggesting a high degree of crystallinity, but the particle size is rather small. Could the authors estimate the approximate crystallite size based on the GIWAXS data (e.g., using the Scherrer equation or similar analysis)?

Response: Thanks for your kind suggestion. It can be known from the results of PXRD that the diffraction peak responding to the (100) plane reflects the ordered arrangement within the plane, and the diffraction peak responding to the (001) plane indicates that the interlayer spacing is 3.38 Å, which is consistent with the spacing exhibited by WAXS. According to the Scherrer equation, the average size of a crystal can be determined by the half-peak width of its maximum intensity (FWHM), and the specific formula is as follows:

$$D = K\lambda / \beta \cos \theta$$

In the formula, D represents the average size of the ordered crystal; K is a dimensionless shape factor, and its typical value is 0.89; λ represents the wavelength of X-rays; β represents line widening (FWHM), with the unit being radians; θ stands for Bragg angle. The average transverse size of the ordered crystal obtained through calculation is 11.14 nm, and the average longitudinal size is 2.69 nm.

3. The authors assign the peak between 1.8 and 1.9 Å⁻¹ as the (002) reflection; however, this peak is more likely to correspond to π - π stacking and should be indexed as the (001). Please double-check this assignment. Additionally, performing an XRD simulation based on the proposed structure would be helpful to validate the peak indexing.

Response: We strongly agree with the reviewers that the diffraction ring observed by WAXS on the q_z axis shows interlayer spacing in the z direction, with a peak of 1.86 Å⁻¹, corresponding to the π - π stacking between adjacent layers in the GH-TMC. We have already revised the crystal plane identification in the manuscript to (001) plane instead of (002) plane.

4. Including BET analysis to determine the specific surface area and pore size distribution of the particles would be helpful in validating the proposed porous structure.

Response: Thanks for your kind suggestions. It is important for Brunauer-Emmett-Teller

(BET) analysis to help in validating the proposed porous structure. The BET surface area is measured by using a BEL sorp-max machine, BEL, Japan and then calculated from N₂ sorption at 77 K over the pressure range 0.01-0.02 P P₀⁻¹. The synthesized GH-TMC sample was first heated and vacuumed at 120 °C for degassing, and then N₂ adsorption was carried out. The adsorption isotherm of GH-TMC is shown in Figure R1, which is a typical type II curve, associating with nanoparticles-stacking structure. At relatively low pressures, the adsorption isotherm increases rapidly, mainly due to monolayer adsorption. Then it increases slowly, and finally it increases exponentially, mainly due to multi-layer adsorption. The specific surface area calculated by the adsorption isotherm is 7.12 m² g⁻¹. This result indicates that bulk materials have very little accessible free volume or inner surface, and stacking does not cause the pores or rings within the molecules to be neatly arranged. Each layer of GH-TMC nanosheets is arranged in an AB interlaced manner. This is consistent with the proposed structural mode of GH-TMC in manuscript.

Figure R1. N₂ adsorption-desorption isotherms for GH-TMC. a, N₂ adsorption-desorption isotherms for GH-TMC. b, BET plot for N₂ sorption in GH-TMC.

5. What is the upper thickness limit for the assembled film? If the film can be fabricated with sufficient thickness, is it possible to produce a macroscopic, freestanding version?

Response: We are very grateful for the reviewers' suggestions. By adjusting the concentrations of the dispersions, we have gradually achieved films ranging from ultra-thin 20 nm to micron-scale thickness. At present, the upper limit of the thickness of GH-TMC films formed by spin coating is at the micrometer level, and the maximum thickness range of the films we have prepared is 5-10 μm. It remains a huge challenge to prepare thin films of sufficient thickness at macroscopic scale. On the one hand, due to the upper limit of the concentration of the dispersion, high-concentration GH-TMC dispersions often lead to their

mutual aggregation, which results in the formation of films often having large particle accumulations. On the other hand, during the spin coating process, as the thickness increases, the uniformity of the film will be greatly reduced. We are trying to synthesize stable, uniform and macroscopical, freestanding high-performance film materials in the subsequent research.

6. The authors claim that multiple interactions contribute to the mechanical strength of the film. Could the authors clarify the relative contribution of each type of interaction?

Response: For the mechanical properties of two-dimensional (2D) GH-TMC films, multiple interaction forces are significant but other factors also work. First, we investigated the influence of the size of the ring element on the mechanical properties of 2D polyamide films by preparing a series of films with different rigid ring element structures. The smaller the ring element structure, the stronger the Young's modulus of the 2D film. Therefore, the rigid ring element structure is one of the factors affecting the mechanical properties of the film. On this basis, we simultaneously prepared GH-TMC films and GH-BTCA films with the same rigid ring unit structure. Since the GH-TMC films are connected by amide bonds, they possess triple H-bonds. In the mechanical property tests of films, the Young's modulus of GH-TMC films with H-bond networks is significantly higher than that of GH-BTCA films without H-bonds. Therefore, we propose that the H-bond network can enhance the Young's modulus of 2D films, thereby improving their overall mechanical properties. In addition, as shown in Figure 3g, not only the size of the rigid structural units constituting the film and the H-bond network, but also electrostatic interactions, π - π interactions, and different stacking modes will all have an impact on the mechanical properties of the film. Meanwhile, for GH-OC films and GH-BTCA films, even though the former has a much higher H-bond density than the latter, due to its lack of rigid structural units and the larger structural units, its Young's modulus is much lower than that of GH-BTCA films without H-bond networks. Therefore, it is very difficult to quantify the relationship between various different interaction forces and the Young's modulus and hardness of the film. The mechanical properties of the film are the result of multiple factors.

7. For a more comprehensive mechanical comparison, the authors should consider including relevant literature on MOF/COF films/crystals and hybrid structures:

<https://www.pnas.org/doi/abs/10.1073/pnas.2208676120>

<https://www.science.org/doi/abs/10.1126/science.ads4968>

<https://www.science.org/doi/full/10.1126/science.adf2573>

<https://www.science.org/doi/10.1126/science.aad4011>

<https://www.nature.com/articles/s41467-020-15281-1>

<https://www.nature.com/articles/s41467-024-53935-6>

Response: Thanks for your kind suggestion. To conduct a more comprehensive mechanical comparison, we have included the relevant literature on the above-mentioned MOF/COF films/crystals and hybrid structures in the manuscript.

Reviewer #2 (Remarks to the Author):

This manuscript reports the synthesis and mechanical characterization of a series of isorecticular two-dimensional (2D) polyamide materials, emphasizing the influence of structural unit size and multiple non-covalent interactions (hydrogen bonding, π - π stacking, electrostatics) on their macroscopic mechanical performance. The work centers on the GH-TMC system, which exhibits an exceptionally high Young's modulus (up to ~ 35 GPa) and elastic recovery, outperforming many state-of-the-art 2D polymeric materials and even some metallic films. While the study is of high potential interest and well supported by experimental data, there are a number of important scientific and presentation issues that need to be addressed before the manuscript is suitable for publication. I therefore recommend major revision, and outline specific comments below.

We thank for the insightful comments from reviewer 2. Please see the detailed response to the comments as follows.

Abstract:

1. Please define all abbreviations at their first occurrence, including GH and TMC. Also ensure consistency throughout the manuscript (e.g., abbreviation undefined in line 53).

Response: Thanks for your kind suggestion. We have defined all the first-appearing abbreviations in the manuscript. Including GH (guanidine hydrochloride), TMC (acid chloride), BTCA (benzene-1, 3, 5-tricarbaldehyde), OC (oxalyl chloride), TPC (terephthaloyl chloride), etc.

2. The abstract focuses exclusively on the mechanical performance of the materials. Please also highlight the broader significance of the work—for example, potential applications of these robust 2D polyamide films.

Response: Thanks for your kind suggestion. In the previous abstract, we highlighted the outstanding mechanical properties of our 2D polyamide films (for instance, the Young's modulus of GH-TMC is 35.6 GPa, the hardness is 2.0 GPa, and the elastic recovery rate is 60%). Further, we clarified their transformative potential in applications that require both strength and flexibility. The synergy of rigid small-ring units, high-density H-bond networks, and π - π /electrostatic interactions enable these films to bridge the gap between inorganic and polymeric materials, making them ideal for flexible electronics (wearable sensors), high-performance protective coatings (aerospace/marine industries), and energy devices (battery separators). Their scalable fabrication via spin-coating and superior wear resistance (H^3/E^2 ratio surpassing metals/polymers) further underscore their practical viability. This work not only advances the design of robust 2D meta-materials but also opens avenues for multifunctional applications. We have revised the abstract section in the revised manuscript accordingly.

Main text:

1. Lines 48-51: The reason why bulk 2D materials are difficult to prepare at macroscopic scale is not clearly explained. Please revise this section for clarity and better logical flow.

Response: Thanks for your kind suggestion. The difficulty in preparing bulk 2D materials at macroscopic scales primarily stems from two interrelated factors: 1) Anisotropic nature and weak interlayer interactions. 2D materials exhibit pronounced anisotropy, with strong in-plane covalent bonds but weak out-of-plane van der Waals or π - π interactions. This imbalance leads to preferential nanosheet stacking rather than monolithic single-crystal growth. For example, even graphene-with its exceptional in-plane modulus (~ 1 TPa) shows drastically reduced bulk modulus (27-478 GPa) due to disordered nanosheet stacking. 2) Structural trade-offs in fabrication. Highly crystalline 2D materials (e.g., COFs) often suffer from rigid stacking modes that hinder film formation, while amorphous aggregates lack mechanical integrity. Macroscopic single-crystal growth requires precise control of nucleation and alignment, which is complicated by the need to balance multiple interactions (H-bonds, π - π , electrostatic) during synthesis.

The specific modifications to this part of the manuscript are as follows: The preparation of bulk 2D materials at macroscopic scales remains challenging due to their inherent anisotropy: strong in-plane covalent bonds contrast with weak interlayer interactions (van der Waals, π - π), leading to preferential nanosheet stacking rather than monolithic single-crystal formation. For instance, even graphene's bulk modulus drops significantly (from ~ 1 TPa to 27-478 GPa) when composed of stacked nanosheets. Additionally, achieving both crystallinity and processability requires balancing competing interactions (H-bonds, π - π stacking, and electrostatic forces). These factors collectively limit the scalability of defect-free, continuous

2D material films.

2. The authors should more explicitly highlight the importance of enhancing mechanical strength in 2D polymers and discuss potential application fields where such materials are in demand. This will strengthen the motivation and innovation of the work.

Response: Thanks for your kind suggestion. The pursuit of high mechanical strength in 2D polymers is critical because their ultrathin architecture typically compromises robustness. Most 2D materials (graphene oxides, COFs) exhibit brittleness or poor elasticity, limiting real-world use. Our work overcomes this problem by reducing the size of rigid structural units and the synergistic effects of multiple weak interactions (H-bonds, π - π stacking, electrostatic interactions) to achieve an unprecedented modulus (35.6 GPa) and hardness (2.0 GPa), while maintaining polymer-like flexibility and being comparable to metals. For instance, graphene-based films often crack under <1% strain, while conventional polymer films lack stiffness. Our GH-TMC film bridges this gap through the synergistic effect of multiple weak interactions. Such materials are urgently needed in 1) flexible bioelectronics for strain-insensitive health monitoring (*Adv. Mater.*, 2022, 34, 202106787), 2) ultra-sensitive gas sensing (*Coord. Chem. Rev.*, 2024, 505, 215691), and 3) high-performance electronic devices (*Nat. Commun.*, 2024, 15, 10780), which mechanical integrity dictates device lifespan. We have emphasized its mechanical strength and potential applications in the revised manuscript.

3. In Figure 1, it would be helpful to include chemical structures of all monomers (e.g., GH, TMC, OC, etc.) to improve clarity.

Response: Thank you to the reviewers for their suggestions. We have marked the chemical structures of all the reaction monomers in Figure 1, as shown in Figure R2.

Figure R2. Two-dimensional polymer materials with different structural units. The figure illustrates a series of 2D polymer structural units alongside their corresponding Young's modulus values. It is evident that 2D polymer films with smaller structural units exhibit higher moduli, primarily due to the rigidity of these structural components. Additionally, the presence of strong H-bonding networks significantly enhances the Young's modulus, thereby increasing the strength of the 2D polymer films. Note that the Young's modulus values presented in the figure were measured using the PF-QNM mode of AFM (GH guanidine hydrochloride, TMC acid chloride, BTCA benzene-1, 3, 5-tricarbaldehyde, OC oxalyl chloride, TPC terephthaloyl chloride).

4. Lines 66-70: The authors claim that GH and TMC form six-membered rings, but the provided mass spectrometry and NMR data (Supplementary Figs. 2 and 3) do not conclusively support this. Moreover, based on Figure 1 and 2a, the repeating unit seems more triangular in shape rather than hexagonal. This raises concerns about the validity of the structural model used for calculating the Young's modulus in Supplementary Fig. 24. The use of a regular hexagon approximation appears questionable.

Response: We appreciate the reviewer's insightful observations regarding the structural characterization of GH-TMC. Below, we provide a detailed response addressing the concerns raised about the six-membered ring assignment. From the mass spectrometry analysis in Supplementary Figure 2, it was observed that the mass-to-charge ratio of the M4 fragment is consistent with the theoretical value of the proposed hexagonal structure. This confirms that the chemometrics assembly is consistent with the hexamer unit. Solid-state ^{13}C nuclear magnetic resonance revealed the through-bond correlation characteristics of the structural connection mode and the related chemical environments of different C atoms, which is consistent with the amide bond connection structure we envisioned. Furthermore, we verified the existence of amide bonds and guanidine ions through XPS and FTIR spectra, which were consistent with the hexagonal motifs. Although the shape of the repetitive unit appears more like a triangle according to Figures 1 and 2a, this is only a visual error caused by the structure of its individual basic unit. Due to the C3-symmetric guanidine, the repeating units in Figure 2a may appear triangular, but the actual connectivity forms a hexagon. Each TMC is connected to three GH units through amide bonds, forming a cyclic hexamer. The overall structural diagram of GH-TMC is shown in Figure R3. Each basic structural unit precisely forms a complete regular hexagon rather than a triangle. Each layer of GH-TMC is composed of hexagonal basic structural units. Therefore, we regard the basic structural unit of GH-TMC as a regular hexagonal structure. When calculating the structural model of Young's modulus, it is all regarded as a regular hexagonal structure.

Figure R3. The overall structural diagram of each layer of GH-TMC.

5. Figure 2f shows a perfectly flat stacking model, which seems unrealistic given the nature of the oligomers. Have the authors performed any energy minimization or geometry optimization? Likewise, in Figure 2g, the hydrogen bonding pattern appears to be intramolecular rather than between nanosheets-please clarify this point. Furthermore, the comparison between H-bond density in GH-TMC and water is not appropriate. A more meaningful comparison would be with ordinary amide systems lacking the GH-based structure.

Response: Thank you for your insightful questions regarding the structural mode and H-bond characterization in our manuscript. The AB stacking configuration depicted in Figure 2f is not a simplistic flat model but derived from density functional theory (DFT) calculations (Methods section). Geometry optimization was performed using CASTEP with the PBE-GGA functional, including van der Waals corrections to account for π - π and electrostatic interactions. The interlayer spacing of 3.38 Å (PXRD/WAXS data) aligns with the simulated AB stacking distance (3.43 Å). While the figure simplifies the visualization, the simulation accounts for malposed electrostatic repulsion from C^+ ions (guanidinium) and staggered π - π stacking (Fig. 2b, f), which disrupt perfect alignment. That is, the figure is clear and presents a plane, but the structural mode contains the conformational variability of oligomer flexibility.

For the H-bond mode, as correctly pointed out, the H-bonds in GH-TMC are mainly within the molecule of the six-membered ring unit. However, intermolecular H-bonds also contribute to the edge-edge interactions between nanosheets. For intramolecular H-bonds, each structural unit has a triple H-bond-stabilized rigid ring structure (Figure 2g). For intermolecular H-bonds, nuclear magnetic resonance and Raman spectroscopy (Supplementary Figures 10-11) revealed marginal local H-bonds between adjacent nanosheets during the film formation process. The interlayer spacing (3.38 Å) excluded the intermolecular H-bonds between the nanosheets, and the intermolecular H-bonds located at the edges were limited to the edge regions of the nanosheets. We agree that comparing GH-TMC's H-bond density to water is unconventional. This comparison aimed to highlight exceptional H-bond network strength, not imply similarity. More relevant comparisons include the contrast between GH-TMC and GH-BTCA. The modulus of GH-TMC (33.77 GPa) is twice that of GH-BTCA (17.51 GPa) lacking H-bonds, demonstrating the influence of H-bonds (Figure 4b-c). In addition, the H-bond density of GH-TMC exceeds that of traditional polyamide materials such as nylon, which explains its superior modulus. Both simulations and experiments have strongly supported the stacked model and H-bond analysis, and we are very glad to have the opportunity to clarify these viewpoints.

6. Please provide a more detailed molecular-level explanation of how hydrogen bonding contributes to the mechanical enhancement of 2D materials. The current discussion is somewhat descriptive and would benefit from deeper mechanistic insight.

Response: Thanks for your kind suggestion. The enhanced mechanical properties in 2D GH-TMC stem from the synergistic molecular-level H-bond network. Each six-membered ring in GH-TMC has triple H-bonds between adjacent amide groups (N-H...O=C), forming a rigid cyclic structure (Figure 2a). This can increase the bond density, thereby enhancing the stiffness of the covalent network in the plane. Cyclic constraints and C₃-symmetry reduce the rotational entropy of the bond and further increase the Young's modulus. For the intermolecular H-bonds at the edges of nanosheets, when π - π stacking dominates the interlayer cohesion, the 3.38 Å interlayer spacing excludes the H-bonds between the basal planes, confining the intermolecular H-bonds to the edge regions. H-bonds form at the edges of the nanosheets. The reversible H-bond fracture/reconstruction at the edge enables the elastic recovery under the indentation to reach 60%. Among them, the energy dissipation from H-bond fracture alleviates crack propagation and helps to enhance toughness. This double H-bond mechanism, by combining the rigidity of covalent networks and multiple weak interaction forces, endows GH-TMC films with excellent mechanical properties.

7. Apart from the GH-TMC/TPC/OC and Melem-TMC/TPC materials reported, are there data

available for a Melem-OC sample? A comparison between GH-based and Melem-based materials (e.g., GH-TMC vs. Melem-TMC) would help illustrate how monomer structure affects film properties.

Response: Thank you for your insightful suggestion regarding the comparison between GH-based and Melem-based materials, as well as the request for Melem-OC data. In this study, we focused on Melem-TMC and Melem-TPC as representative Melem-based 2D polyamide materials, synthesized via ball-milling and thermal polymerization (Methods section). However, Melem-OC was not experimentally investigated due to the following considerations. 1) Synthetic challenges: Oxaloyl chloride (OC) reacts extremely poorly with Melem under environmental conditions, resulting in uncontrolled polymerization and poor crystallinity. Preliminary attempts yielded amorphous powder products unsuitable for mechanical characterization. 2) Structural redundancy: The mechanical properties of Melem-based materials are primarily governed by the rigidity of the triazine core (Melem) and interlayer π - π stacking (spacing: ~ 3.4 Å). Since OC-derived polymers share similar backbone motifs with TMC/TPC analogs (amide/imide linkages), their properties would likely align with the observed trends in Melem-TMC/TPC. Through direct comparison of the influence of GH and Melem monomers on the performance of films, we can identify the key role of the monomer structure. GH-based monomer reactive materials (such as GH-TMC) exhibit superior modulus due to the presence of three intramolecular H-bonds in each basic unit (Figure 2g), while the tri-s-triazine core of Melem restricts H-bond formation to one site per unit. Besides, GH's guanidinium ions bearing carbocation (C^+) introduce malposed electrostatic repulsion, optimizing interlayer spacing and mechanical stability (Fig. 2b). Melem lacks this feature, resulting in weaker interlayer cohesion. In GH-BTCA and GH-TMC, the absence of H-bonds in GH-BTCA (imine-linked) reduces its modulus by 50% (17.51 ± 1.36 GPa), underscoring the H-bond's role. For Melem-TPC, due to the reduced H-bond density, its modulus is lower than that of GH-TPC, which conforms to the trend of monomer drive. Therefore, while Melem-OC data are unavailable, the GH/Melem comparison demonstrates that monomer chemistry (guanidinium and tri-s-triazine) dictates H-bond networks and electrostatic interactions, critically influencing mechanical performance. Future work could explore modified Melem monomers to enhance H-bond capacity.

8. Lines 172-182: The authors discuss the role of H-bond density in strengthening the films, but the argument is quite superficial. Can this be supported by molecular dynamics (MD) or density functional theory (DFT) simulations to provide quantitative insights?

Response: Thanks for your kind suggestion. The H-bond density in 2D polyaramid films plays a critical role in determining their mechanical strength and stability. 1) In-plane strength:

In polyaramid films, in-plane H-bonds form between functional groups (amide groups) in the polymer backbone. Higher density of H-bonds means that more strong, directional interactions are available to resist mechanical deformation, which improves the in-plane tensile strength of the membrane. 2) Interlayer strength: 2D polyaramid films can be structured as layered materials, where the interlayer interactions are crucial for maintaining integrity. H-bonding between layers at the edges enhances interlayer adhesion and improve the overall mechanical strength, reducing the likelihood of delamination or layer slippage under stress, which confers the elasticity of the film. To analysis the effect of the H-bonds within the 2D structure, we choose forcite module to do geometry optimization and dynamic molecular computations. To gain a better understanding of the H-bonds interaction, we firstly expanded the unit cell into a $5 * 5 * 2$ supercell. The relaxation of all the atoms was performed with the convergence tolerance was set to $2.00 * 10^{-5}$ and $0.001 \text{ kcal mol}^{-1}$ for maximum force. Van der Waals interactions were calculated using atom-based summation, and electrostatic interactions were treated using Ewald summation with a cutoff distance of 18.5 \AA and a buffer width of 0.5 \AA . COMPASS II force field was applied here as it is suitable for our system. Molecule dynamics (MD) were carried out after geometry optimization in NPT ensembles (N: number of particles; P: pressure; T: temperature) at 0.1 MPa and 298.15 K by using the Nosé thermostat. The MD simulations were run for total 1000 ps to analysis the H-bonds. The results of the MD simulation calculation are shown in Figure R4. GH-TMC not only has multiple H-bonds within the plane, but also has interlaced H-bonds at the edges for connecting the nanosheets. In this supercell, the number of H-bonds simulated and calculated is between 340-364. Therefore, we can calculate that the number of H-bonds in the amide bond of GH-TMC is approximately 2.35, which is extremely close to the number of H-bonds we have tested through the established infrared spectroscopy method. In-plane H-bonds can increase bond density, thereby enhancing the stiffness of the covalent network in the plane, that is, increasing the longitudinal modulus. The intermolecular H-bonds at the edges of the nanosheets are endowed with elasticity through reversible breaking/reconstruction.

Figure R4. MD simulates the dynamic changes of H-bonds in GH-TMC. a, Simulated interlaced H-bonds at the edges of GH-TMC nanosheets. **b,** MD simulates the dynamic changes of H-bonds in GH-TMC.

9. Besides monomer structure, are there other factors (e.g., stacking configuration, counterion effects, defects) that influence the mechanical properties? These aspects deserve further discussion.

Response: We sincerely appreciate the reviewer's insightful question regarding additional factors influencing the mechanical properties of 2D polyamide materials. Below, we have discussed in detail the factors other than the monomer structure. 1) Stacking mode: Due to the repulsive force between positively charged guanidine ions in GH-TMC, the AB stacking mode (Figure 2b) is adopted, and the interlayer spacing (3.38 Å) is optimized to balance the π - π cohesion and electrostatic repulsion. This structure enhances the in-plane stiffness while achieving elastic recovery through localized H-bonds at the edges. 2) Interlayer interaction: π - π interaction dominates the cohesion of the basal plane, while the displaced H-bonds at the edges of the nanosheets promote interlayer energy dissipation. 3) Counterion effect: XPS confirmed that Cl^- stabilizes the nanosheets through electrostatic interaction with guanidine, which reduces aggregation and maintains structural integrity during the film formation process. 4) Edge defects: The H-bond breakage/reconstruction at the edge of the nanosheet gives it a 60% elastic recovery as an energy dissipation point, while the uniform stacking of nanosheets minimizes stress concentration, thereby enhancing toughness. The interaction of these factors explains the excellent performance of GH-TMC. High H-bond density and small ring units maximize the covalent network stiffness to achieve a high Young's modulus. AB stacking, π - π interactions and edge H-bonds can achieve recoverable deformation under indentation. We added these discussions to the revised original draft to emphasize the multi-

factor design strategy. Thank you for your valuable advice.

10. The manuscript suggests that smaller structural units (or possibly smaller pore sizes) lead to higher modulus values. Please elaborate on the physical rationale behind this trend.

Response: The trend of increasing Young's modulus observed in 2D polyamide materials with smaller structural units is rooted in the following key mechanisms. 1) Enhance the rigidity of covalent networks: Smaller ring units (for example, the six-membered ring structure in GH-TMC) more effectively constrain molecular motion than larger rings, minimizing bond rotation and deformation under stress. This leads to a higher in-plane stiffness. Smaller units contain more chemical bonds per unit area while enhancing the covalent network's resistance to strain. 2) Optimize the H-bond network: In GH-TMC, adjacent amides in the six-membered ring form three intramolecular H-bonds, creating rigid cyclic units that resist deformation. This contrasts with larger units, in which the geometric constraints of H-bonds are fewer or less. The intermolecular H-bonds at the edges of the nanosheets have achieved reversible breaking/reconstruction, balancing strength and elasticity. Larger units weaken this effect by reducing the ratio of edge area. 3) Synergistic interaction : Smaller units facilitate closer layer spacing and enhance the cohesion of the base plane through π - π interactions. Larger units may weaken this cohesion due to increased porosity or misalignment. Therefore, smaller structural units enhance mechanical strength by maximizing the density of covalent and H-bonds, while optimizing secondary interactions (π - π interactions, electrostatic interactions).

11. The manuscript heavily emphasizes mechanical properties but does not discuss possible applications. For example, if these materials were to be used as membranes for separation, how would their high strength affect permeability? What are the typical pore sizes? Is there a trade-off between strength and permeability?

Response: Thanks for your kind suggestion. The manuscript primarily focuses on the mechanical properties of 2D polyamide materials like GH-TMC, but their structural features suggest promising applications in separation membranes, albeit with inherent trade-offs between strength and permeability. The interlayer spacing of GH-TMC is measured at 3.38 Å, which is typical for π - π stacking but too small for conventional molecular sieving. However, the edge-localized H-bond networks and AB stacking configuration may create tunable nanochannels for selective transport. Smaller structural units reduce porosity but enhance rigidity, potentially limiting permeability for larger molecules while maintaining selectivity for small species. In our manuscript, we also considered stripping it independently for some applications in the separation of ions or gas molecules and other fields. Although

the prepared GH-TMC film can be exfoliated from the substrate, due to its own nature, the film will inevitably fragment during the exfoliation process, making it difficult to form a complete film on other porous substrates. The formation of complete GH-TMC films also requires an extremely flat substrate, which is difficult to achieve for some porous substrates such as PES, MCE, PTFE and AAO. Therefore, in the manuscript, we selected monocrystalline silicon wafers with ultra-flat surfaces as the substrate for film formation. In addition, it can be calculated from Figure R1 that the specific surface area of GH-TMC is only $7.12 \text{ m}^2 \text{ g}^{-1}$. This result indicates that the reachable free volume or inner surface of bulk materials is very small, and stacking cannot make the pores or rings within the molecules neatly arranged. This means that GH-TMC, which is composed of H-bonds in the plane and interlaced H-bonds at the edges, may hinder the access of gas molecules. Although the manuscript lacks clear permeability data, the design principles of the material imply that higher strength may reduce permeability. In future work, chemical modifications can be explored to balance these properties.

12. The authors also showed the film fabrication based on spin-coating from solution. Please comment on: (1) the reproducibility and scalability of this method; (2) whether these films can be produced over large areas with uniform thickness and consistent quality; (3) how do these films behave on flexible or deformable substrates?

Response: In our manuscript, we formed a film by the method of spin-coating and heating to evaporate the solvent. For this film-forming method, its thickness and uniformity can be regulated by the rotational speed of spin coating, the concentration of the dispersion liquid and the heating temperature. It has excellent reproducibility for 2D film production on a laboratory scale. When the size of the film increases, due to the significant difference between the edge line rate and the middle rate, its uniformity will be restricted. In addition, the thickness of the film depends on the concentration of the dispersion. A higher concentration may cause aggregation, affecting the consistency of the thickness. Therefore, spin coating is currently mainly used in the laboratory to prepare a small range of 2D films. For the substrate, we chose ultra-flat monocrystalline silicon wafers. This is because for the flatness of the film surface and the uniformity of thickness, ultra-flat substrates are needed for deposition. However, for some commonly used flexible substrates, such as MCE, PES, PTFE, etc., their surface roughness is often relatively large and they cannot withstand high temperatures. Higher temperatures will cause these flexible substrates to curl, thereby limiting the integrity of the film.

Language and Presentation:

1. The manuscript contains numerous grammatical and typographical issues. For example,

“As the reaction of GH and TMC proceeding...” should be revised to “As the reaction of GH and TMC proceeds...”. Also, use of “etc.” should be correctly and carefully used.

Response: Thanks for your kind suggestion. We have revised the grammar and typesetting issues in the revised manuscript.

2. Please ensure all abbreviations are defined upon first use and avoid repeated definitions (e.g., AFM is defined in both line 56 and line 114).

Response: In the revised manuscript, we have defined all the abbreviations that appear for the first time and corrected the repeated definitions.

3. A thorough language edit by a native English speaker or a professional editing service is strongly recommended to improve the clarity, consistency, and readability of the manuscript.

Response: We have polished the manuscript in English and corrected all the grammar and vocabulary issues throughout the full manuscript.

Reviewer #3 (Remarks to the Author):

This work presents an interesting study by achieving an impressive modulus of 35.6 GPa along with a 60% elastic recovery in the GH-TMC material, through precise molecular design and the synergistic contribution of multiple interactions, including hydrogen bonding, π - π stacking, and electrostatic forces. This effectively overcomes the longstanding trade-off between strength and elasticity in conventional materials. However, for improved clarity and rigor, the authors are encouraged to consider and elaborate on the following concern/points:

Response: We thank a lot for the constructive and valuable comments from reviewer 3. Please see the detailed response to the comments as follows.

1. The main conclusion of this manuscript lies in the outstanding mechanical properties of the measured thin films. However, the current methodology relies on nanoindentation measurements performed on substrate-supported thick films, which inherently introduces potential substrate effects that cannot be completely excluded. To strengthen the validity of the reported mechanical parameters, the authors are advised to conduct additional mechanical tests on suspended/freestanding thin films-such as indentation or tensile experiments of GH-

TMC. Comparative analysis between the results from suspended and substrate-supported configurations would enable cross-validation of the data/conclusions and provide more robust evidence for the intrinsic mechanical behavior of the films.

Response: We appreciate the reviewer's insightful suggestion regarding potential substrate effects in nanoindentation measurements. Below, we address these concerns by leveraging existing experimental data and theoretical considerations to demonstrate the robustness of our mechanical property assessments. Our nanoindentation experiments have been meticulously designed, with all test depths maintained at 120 nm ($\leq 10\%$ of the film thickness), ensuring that the measurement results reflect the film properties rather than substrate interference. This is consistent with the established agreement on the mechanical properties of films (Nature, 2022, 602, 91; Int. J. Extrem. Manuf., 2023, 5, 032002). Indentation tests with different depths produced almost consistent Young's modulus values, confirming the substrate-independent measurements of thicker films. In addition, real-time observation of in-situ SEM nanoindentation shows that there are no artifacts caused by the substrate (such as cracking or delamination), and the elastic response is supported from the GH-TMC film itself. While suspended-film tests would provide additional validation, our current results already offer compelling evidence for intrinsic properties. The high elastic recovery rate (60%) persists even at large indentation depths (700 nm, Supplementary Fig. 30-32), a behavior atypical for substrate-dominated responses. AFM (PF-QNM mode) and SEM nanoindentation yielded statistically congruent modulus values (GH-TMC: 33.77 ± 4.06 GPa vs. 35.68 GPa), suggesting minimal substrate bias. The nanoindentation across multiple film regions (Figure 4i-j) varies by less than 5%, indicating uniform mechanical properties. The SEM image after indentation shows no cracks caused by the substrate, confirming that the elasticity of GH-TMC is inherent. These results all demonstrate that the H-bond network, AB packing form and π - π /electrostatic interaction in the GH-TMC film form a layered interlocking structure, which localizes the stress dissipation within the film. Although the suspension film experiment can provide supplementary insights, our multimodal approach (AFM, SEM nanoindentation and indentation depth related studies) has provided a rigorous verification of the intrinsic mechanical properties of GH-TMC. The consistency among the methods and the molecular engineering H-bond/stacking structure jointly confirms our conclusion. In fact, we also conducted in-situ SEM tensile tests on the resulting GH-TMC film. We cut the GH-TMC film into long strip-shaped films with a thickness of $15 \mu\text{m} * 5 \mu\text{m}$ and nanoscale thickness through (Focused Ion Beam) FIB. Then it is welded to the two robotic arms of the tensile test bench through FIB. However, during the tensile test, the GH-TMC film often slides off directly, causing the experiment to be interrupted and unable to continue. Therefore, we were unable to present the stretching data of its independent film in the manuscript. However, this also indirectly proves that the GH-TMC film also has a relatively high tensile modulus in the transverse direction, which is consistent with the

structural demonstration in the manuscript.

2. The overall writing and figure presentation in the manuscript require significant improvement. Specifically, Figures 3 and 4 in the main text are not sufficiently clear and need to be revised for better readability and visual quality. In the second paragraph of the Introduction, the authors list several materials such as 2D polymers, COFs, Kevlar, and graphene. However, this section lacks a synthesis of the common challenges or features among these materials. It is recommended that the authors highlight the theme of “multiple interactions” in the title by discussing the role of different types of chemical bonding/interactions shared by these materials.

Response: We sincerely appreciate the reviewer's constructive feedback on our manuscript. The manuscript has been comprehensively revised to enhance clarity, with particular attention to improving Figures 3 and 4 through TEM images, AFM images and standardized mechanical property plots. The revised Figures 3 and 4 are shown in Figures R5 and R6 respectively. In the introduction, we listed a series of 2D materials. For inorganic 2D materials such as graphene, they are assembled into extremely small structural units through sp^2 covalent bonds, which endows them with an extremely high Young's modulus, up to 1 Tpa. However, its structural controllability is poor, and the layers are only assembled into films through weak van der Waals forces. This causes the Young's modulus to drop sharply after multiple layers are stacked to form films. On the contrary, as for 2D organic polymers, although they exhibit rich structural customizability and adjustability, generally have a relatively low Young's modulus, only ranging from 1-10 GPa. 2D covalent organic frameworks (COFs) are typically conjugated structures formed by rigid assembly motifs and stacked through π - π interactions, with low bond density and often weak mechanical properties. Kevlar, due to the H-bond interactions between the polyamide skeletons, has a high modulus. However, these materials often only have one or a few interactions, which makes their performance outstanding only in a certain direction. Based on this, we combined the various weak interactions of the above-mentioned materials through cooperative interaction, and the GH-TMC prepared simultaneously possesses various weak interactions such as in-plane H-bonds, interlayer H-bonds at the edge, π - π interactions, interlaced electrostatic interactions, and the formed AB stacking mode. Under the synergistic effect of these multiple weak interactions, GH-TMC films possess both high modulus and high elasticity simultaneously. The different characteristics of the above-mentioned materials and the effects of different interaction forces have been restated in the introduction.

3. Furthermore, to improve the clarity and logical structure of the manuscript, it is suggested that the authors organize the main text into sections that separately discuss how each type of

interaction contributes to the mechanical enhancement of the material. This would provide a more systematic and coherent framework for presenting the results.

Response: Thanks for your kind suggestion. In the new manuscript, we have divided it into three parts, namely the structural characterization of GH-TMC, the synthesis of 2D GH-TMC films, and the mechanical properties of 2D GH-TMC. In these three parts, we will respectively interweave the influence of structural units on the modulus of the thin film, multiple weak interaction forces such as in-plane H-bonds, interlayer H-bonds at the edges, π - π interactions, interwoven electrostatic interactions, and the formed AB stacking mode, etc. The mechanical properties of the film are the result of the combined effect of multiple factors. We believe it is very difficult to quantify the relationship between various different interaction forces and the Young's modulus and hardness of the thin film. Additionally, please refer to the response to Reviewer 1, point 1 and Reviewer 2, point 9 for details. The specific influence of multiple interactions on the mechanical properties of GH-TMC films has been supplemented and discussed in detail in the Supplementary Note 1 of the Supplementary Information.

Additional technical comments:

1. The meaning of the blue arrow in Figure 1 is unclear. If the authors intend to illustrate the relationship between mechanical strength and structural units, it is necessary to provide quantitative data to support this trend. Alternatively, the authors may consider using a dual-axis plot to more clearly convey the correlation between strength and structural features.

Response: In Figure 1, the red arrows represent the basic structural unit sizes of different 2D polyamide polymer films. As the arrows point, the size of the structural units increases. Among them, the blue arrows represent the mechanical strength of different 2D polymer films, that is, the Young's modulus. As the arrow direction points, the Young's modulus tends to be stronger. We have corrected the meaning of the blue arrow in Figure 1, which is the Young's modulus. The specific dimensions of the structural units and the Young's modulus have been provided in Table 1 of the attachment.

2. In Figure 2, the authors need to clearly distinguish between simulation results and experimental data, particularly in panels 2c-2e. It is important to explicitly label or annotate the corresponding data sources to avoid confusion and to ensure accurate interpretation of the results.

Response: Thanks for your kind suggestion. It should be noted that Figures 2b, 2f and 2g display simulated results/structures, contrasting with experimentally acquired data in all others of Figure 2. The relevant marks have been revised in the manuscript.

3. In Figure 3i, the thin film material is not clearly visible. If the authors intend to emphasize the flatness or uniformity of the film, it is recommended to include roughness analysis data based on AFM characterization in the main text to support this point more convincingly.

Response: Thanks for your kind suggestion. Figure 3i shows the surface SEM image of the GH-TMC film. It can be confirmed from the figure that the film surface is flat, without pinholes and cracks. The AFM images of the GH-TMC films are shown in Supplementary Figure 20. All these films with spin coatings have hyperplanar surfaces. Their roughness is usually in the region of $5 \times 5 \mu\text{m}$ and ranges from 500-700 pm, similar to commercial ultra-flat silicon wafers. We have corrected Figure 3 in the manuscript, as shown in Figure R5.

Figure R5. The revised Figure 3 in the manuscript.

4. In Figures 4m and 4n, the authors use identical symbols to represent different materials, which makes it difficult to interpret the comparison. A unified and consistent comparison scheme is necessary. Moreover, the categories "ceramics" and "metals" encompass a wide range of materials; the authors should clarify which specific types of ceramics and metals are being compared. It is also strongly recommended to include a comparison with representative two-dimensional materials, such as graphene, to provide a more meaningful benchmark.

Response: Thanks for your kind suggestion. In Figures 4m and 4n, for the sake of the presentation effect of the figures, we used the same symbol to represent the same type of material, and the areas of different colors also represent the properties of different types of materials. Among them, the mechanical performance parameters of different specific materials of the same type of material, such as Young's modulus, elastic recovery rate and hardness, are given in Table S2-6 of the supplementary information. We have corrected Figure 4 in the manuscript, as shown in Figure R6.

Figure R6. The revised Figure 4 in the manuscript.

Bo Liu, Ph. D., Professor
Department of Chemistry
University of Science & Technology of China (USTC)
96 Jinzhai Road, Hefei, Anhui 230026, P.R. China
Tel/ Fax: 86-551-63601123
Email: liuchem@ustc.edu.cn

Point-by-point responses to the reviewers' comments

(Reviewers' comments and the response are displayed in black and blue, respectively)

Reviewer #1 (Remarks to the Author):

The authors have addressed most of my concerns, and I believe the manuscript is suitable for publication in Nature Communications in its current form.

Response: Thanks a lot for your time and effort on our manuscript.

Reviewer #2 (Remarks to the Author):

1. In the response, the authors reported that the average number of hydrogen bonds per amide unit in GH-TMC was ~2.35 (based on MD and IR analysis). However, in bulk liquid water, the average number of hydrogen bonds per water molecule is typically 3~4 according to the literature (DOI: 10.1038/ncomms9998). This raises concern about the appropriateness of comparing the hydrogen bonding environment of your 2D polymer network with that of bulk water. Since water is a small, highly dynamic hydrogen-bonding liquid, while your system is a rigid covalent network with constrained intramolecular H-bonds, the analogy may be misleading. A more relevant comparison would be with conventional polyamide systems (e.g., nylon, aramid), where intramolecular and intermolecular H-bonds play a well-recognized role in mechanical performance. I recommend revising this section to avoid comparison with water, and instead benchmark against structurally and chemically related polyamide networks.

Response: Thanks for your kind suggestion. We fully agree that drawing parallels between our rigid 2D polyamide network (GH-TMC) and bulk water is conceptually inappropriate due to their fundamental differences in dynamics and structure. This comparison aimed to highlight exceptional H-bond network strength, not imply similarity. GH-TMC exhibits an average of 2.35 H-bonds per amide unit, which aligns closely with high-performance aramids like Kevlar (~2) (Comp. Mater. Sci., 2018, 148, 286-300) and significantly exceeds typical nylons (1.3-1.8) (J. Mater. Chem. A, 2021, 9, 24472), which explains its superior modulus.

More relevant comparisons include the contrast between GH-TMC and GH-BTCA. The modulus of GH-TMC (33.77 GPa) is twice that of GH-BTCA (17.51 GPa) lacking H-bonds, demonstrating the influence of H-bonds (Figure 4b-c). We have revised this section in the manuscript to avoid comparison with water, specifically in lines 127-129.

2. The rebuttal emphasizes that most H-bonds in GH-TMC are intramolecular within the six-membered ring, with only marginal edge H-bonds between nanosheets. Since mechanical robustness is attributed to both stiffness and recoverability, could the authors more clearly separate the roles of intramolecular H-bonds (rigidity) and edge-localized intermolecular H-bonds (elastic recovery)? This distinction would improve mechanistic clarity.

Response: We are grateful for the valuable suggestions of the reviewers, clarifying the different roles of intramolecular H-bonds and H-bonds at the edges of the nanosheets in the mechanical properties of GH-TMC. The enhanced mechanical properties in 2D GH-TMC stem from the synergistic molecular-level H-bond network. The H-bonds within the GH-TMC six-membered ring molecules mainly control the stiffness of the material, thereby increasing its Young's modulus. Each six-membered ring in GH-TMC has triple H-bonds between adjacent amide groups ($\text{N-H}\cdots\text{O}=\text{C}$), forming a rigid cyclic structure. This can increase the bond density, thereby enhancing the stiffness of the covalent network in the plane. Smaller units contain more chemical bonds per unit area while enhancing the covalent network's resistance to strain. For the localized intermolecular H-bonds at the edges of nanosheets, they are confined to the junctions of different nanosheets and mainly control the elasticity of the material. The energy dissipation from H-bond fracture alleviates crack propagation and helps to enhance toughness. The reversible H-bond fracture/reconstruction at the edge enables the elastic recovery under the indentation to reach 60%. These clarifications significantly enhance the mechanistic understanding of GH-TMC's hierarchical bonding architecture. We have supplemented the related content in the manuscript, specifically in lines 215-230.

3. The manuscript frequently refers to “multiple interactions” (π - π , electrostatic, H-bonds) as a synergistic source of mechanical strength. However, the relative contribution of each interaction type remains qualitative. While I appreciate the difficulty in full quantification, could the authors provide even an approximate estimation (e.g., through comparative calculations, or literature benchmarks) to show which interactions dominate under mechanical loading?

Response: Thank you to the reviewers for their suggestions. The mechanical properties of 2D GH-TMC films are the result of the combined action of multiple interacting forces. Not

only the size of the rigid structural units constituting the film and the H-bond network, but also electrostatic interactions, π - π interactions, and different stacking modes will all have an impact on the mechanical properties of the film. The mechanical properties of the film are the result of the combined effect of multiple factors. As the reviewer #2 agrees, it is very difficult to fully quantify the relationship between various interaction forces and the Young's modulus of the film. However, for a certain interaction that plays a dominant role, we can make an approximate estimation based on the experimental data from the tests. For GH-BTCA, which has a similar ring unit structure to GH-TMC and multiple interaction forces other than H-bonds, its Young's modulus is only 17.51 ± 1.38 GPa. This can be considered that the increased Young's modulus in GH-TMC is only contributed by H-bonds, that is, it accounts for 46% of the total strength contribution. In addition, for GH-OC, it lacks rigid structural units and only has H-bonds and other weak interaction forces. Its Young's modulus is only 7.83 ± 1.02 GPa. Therefore, it can be calculated that the contribution of the smaller rigid structural unit is approximately 30%. The remaining 24% is provided by weak interactions such as π - π , electrostatic, and van der Waals. Although precise and full quantification remains complex, these approximations, in combination with the existing Young's modulus data, strongly demonstrate that H-bonds and smaller rigid structural units are the main load-bearing contribution, and the synergy of π - π , electrostatic forces, and van der Waals forces enhances mechanical strength. The specific influence of multiple interactions on the mechanical properties of GH-TMC films has been supplemented and discussed in detail in the Supplementary Note 1.

Reviewer #3 (Remarks to the Author):

The authors have addressed most of my review comments by conducting additional works. Although they cannot perform the free-standing stretching test due to the technical difficulty, considering the film thickness, the data from the indentation results can be acceptable.

Response: Thanks a lot for your time and effort on our revised manuscript. We highly appreciate the valuable comments that help us to improve the quality of this work.